# DIFFUSION MODELS WITH LEARNED ADAPTIVE NOISE

## ABSTRACT

Diffusion models have gained traction as powerful algorithms for synthesizing high-quality images. Central to these algorithms is the diffusion process, which maps data to noise according to equations inspired by thermodynamics, and which can significantly impact performance. In this work, we explore whether a diffusion process can be learned from data. We propose multivariate learned adaptive noise (MuLAN), a learned diffusion process that applies Gaussian noise at different rates across an image. Our method consists of three components—a multivariate noise schedule, instance-conditional diffusion, and auxiliary variables—which ensure that the learning objective is no longer invariant to the choice of noise schedule as in previous works. Our work is grounded in Bayesian inference and casts the learned diffusion process as an approximate variational posterior that yields a tighter lower bound on marginal likelihood. Empirically, MuLAN significantly improves likelihood estimation on CIFAR10 and ImageNet, and achieves ∼**2x** faster convergence to state-of-the-art performance compared to classical diffusion.

## 1 INTRODUCTION

Diffusion models, inspired by the physics of heat diffusion, have gained traction as powerful tools for generative modeling, capable of synthesizing realistic, high-quality images (Sohl-Dickstein et al., 2015; Ho et al., 2020; Rombach et al., 2021). Central to these algorithms is the diffusion process, a gradual mapping of clean images into white noise. The reverse of this mapping defines the data-generating process we seek to learn—hence, its choice can significantly impact performance (Kingma & Gao, 2023). The conventional approach involves adopting a diffusion process derived from the laws of thermodynamics, which, albeit simple and principled, may be suboptimal due to its lack of adaptability to the variations within the dataset.

In this study, we investigate whether the notion of diffusion can be instead *learned from data*. We focus on classical Gaussian diffusion, and propose a method for learning the schedule by which Gaussian noise is applied to different parts of an image. Our approach is grounded in Bayesian inference (Kingma & Welling, 2013), allowing the diffusion process to be viewed as an approximate variational posterior—learning this process induces a tighter lower bound on the marginal likelihood of the data. More generally, this methodology allows us to adapt the noise schedule to the specific characteristics of each image instance, thereby enhancing the performance of the diffusion model.

Specifically, we propose a new diffusion process, multivariate learned adaptive noise (MuLAN), which augments classical diffusion models (Sohl-Dickstein et al., 2015; Kingma et al., 2021) with three innovations: a per-pixel polynomial noise schedule, a conditional noising process, and auxiliary-variable reverse diffusion. While previous work argued that the learning objective of a diffusion model is invariant to the choice of diffusion process (Kingma et al., 2021; Kingma & Gao, 2023), we show that this claim is only true for the simplest types of univariate Gaussian noise: we identify a broader class of noising processes whose optimization yields significant performance gains.

Our learned diffusion process yields improved log-likelihood estimates on two standard image datasets, CIFAR10 and ImageNet. Remarkably, we achieve state-of-the-art performance for diffusion models with less than half of the training time of previous methods. Our method also does not require any modifications to the underlying UNet architecture, making it compatible with most existing diffusion algorithms.

**Contributions**   In summary, our paper (1) provides a theoretical argument that suggests that creating a noise schedule for each input dimension and conditioning it on the input yields improved likelihood estimation. We then (2) introduce a novel approach for conditioning the noise schedule on the input via a latent distribution. We demonstrate its effectiveness over conditioning the schedule on the input directly through both empirical and theoretical means. Lastly, we (3) empirically demonstrate that learning the diffusion process leads to $\sim$**2x** faster convergence towards state-of-the-art performance on CIFAR-10 and ImageNet datasets.

## 2   BACKGROUND

A diffusion process $q$ transforms an input datapoint denoted by $\mathbf{x}_0$ and sampled from a distribution $q(\mathbf{x}_0)$ into a sequence of noisy data instances $\mathbf{x}_t$ for $t \in [0, 1]$ by progressively adding Gaussian noise of increasing magnitude. (Sohl-Dickstein et al., 2015; Ho et al., 2020; Song et al., 2020). The marginal distribution of each latent is defined by $q(\mathbf{x}_t|\mathbf{x}_0) = \mathcal{N}(\mathbf{x}_t; \alpha_t \mathbf{x}_0, \sigma_t \boldsymbol{I})$ where the diffusion parameters $\alpha_t, \sigma_t \in \mathbb{R}^+$ implicitly define a noise schedule as a function of $t$, such that $\nu(t) = \alpha_t^2/\sigma_t^2$ is a monotonically decreasing function in $t$. Given any discretization of time into $T$ timesteps of width $1/T$, we define $t(i) = i/T$ and $s(i) = (i-1)/T$ and we use $\mathbf{x}_{0:1}$ to denote the subset of variables associated with these timesteps; the forward process $q$ can be shown to factorize into a Markov chain $q(\mathbf{x}_{0:1}) = q(\mathbf{x}_0) \left( \prod_{i=1}^{T} q(\mathbf{x}_{t(i)}|\mathbf{x}_{s(i)}) \right)$.

The diffusion model $p_{\theta\theta}$ is defined by a neural network (with parameters $\theta$) used to denoise the forward process $q$. Given a discretization of time into $T$ steps, $p$ factorizes as $p_\theta(\mathbf{x}_{0:1}) = p_\theta(\mathbf{x}_1) \prod_{i=1}^{T} p_\theta(\mathbf{x}_{s(i)}|\mathbf{x}_{t(i)})$. We treat the $\mathbf{x}_t$ for $t > 0$ as latent variables and fit $p_\theta$ by maximizing the evidence lower bound (ELBO) on the marginal log-likelihood given by:

$$\log p_\theta(\mathbf{x}_0) = \text{ELBO}(p_\theta, q) + D_{\text{KL}}[q(\mathbf{x}_{t(1):t(T)}|\mathbf{x}_0)\|p_\theta(\mathbf{x}_{t(1):t(T)}|\mathbf{x}_0)] \geq \text{ELBO}(p_\theta, q) \qquad (1)$$

$$= \mathbb{E}\left[\log p_\theta(\mathbf{x}_0|\mathbf{x}_{t(1)}) - \sum_{i=2}^{T} D_{\text{KL}}[q_\phi(\mathbf{x}_{s(i)}|\mathbf{x}_t, \mathbf{x}_0)\|p_\theta(\mathbf{x}_{s(i)}|\mathbf{x}_{t(i)})] - D_{\text{KL}}[p_\theta(\mathbf{x}_1)\|q_\phi(\mathbf{x}_1|\mathbf{x}_0)]\right]$$

In most works, the noise schedule, as defined by $\nu(t)$, is either fixed or treated as a hyper-parameter (Ho et al., 2020; Chen, 2023; Hoogeboom et al., 2023). Chen (2023); Hoogeboom et al. (2023) show that the noise schedule can have a significant impact on sample quality. Kingma et al. (2021) consider learning $\nu(t)$, but argue that the KL divergence terms in the ELBO are invariant to the choice of function $\nu$, except for the initial values $\nu(0), \nu(1)$, and they set these values to hand-specified constants in their experiments. They only consider learning $\nu$ for the purpose of minimizing the variance of the gradient of the ELBO. In this work, we show that the ELBO is not invariant to more complex forward processes.

## 3   DIFFUSION MODELS WITH MULTIVARIATE LEARNED ADAPTIVE NOISE

A standard diffusion process is defined via hand-crafted equations inspired by thermodynamics (Sohl-Dickstein et al., 2015). Here, we investigate whether the notion of diffusion can be instead learned from data. Specifically, we introduce a new diffusion process, multivariate learned adaptive noise (MuLAN), which introduces three innovations: a per-pixel polynomial noise schedule, a conditional noising process, and auxiliary-variable reverse diffusion. We describe these below.

### 3.1   WHY LEARNED DIFFUSION?

Perhaps the most direct motivation for our work comes from Bayesian inference. Notice that the gap between the evidence lower bound $\text{ELBO}(p, q)$ and the marginal log-likelihood (MLL) in Equation 1 is precisely the KL divergence $D_{\text{KL}}[q(\mathbf{x}_{t(1):t(T)}|\mathbf{x}_0)\|p_\theta(\mathbf{x}_{t(1):t(T)}|\mathbf{x}_0)]$ between the diffusion process $q$ over the latents $\mathbf{x}_t$ and the true posterior of the diffusion model. The diffusion process takes the role of an approximate variational posterior in $\text{ELBO}(p, q)$.

This observation suggests that the ELBO can be made tighter by choosing a diffusion processes $q$ that is closer to the true posterior $p_\theta(\mathbf{x}_{t(1):t(T)}|\mathbf{x}_0)$; this in turn brings the learning objective of

closer to $\log p(\mathbf{x})$, which is often the ideal objective that we wish to optimize. In fact, the key idea of variational inference is to optimize $\max_{q \in \mathcal{Q}} \text{ELBO}(p, q)$ over an family of approximate posteriors $\mathcal{Q}$ to induce an tighter ELBO (Kingma & Welling, 2013). Most diffusion algorithms, however optimize $\max_{p \in \mathcal{P}} \text{ELBO}(p, q)$ within some family $\mathcal{P}$ with a fixed $q$. Our work seeks to jointly optimize $\max_{p \in \mathcal{P}, q \in \mathcal{Q}} \text{ELBO}(p, q)$; we will show in our experiments that this improves the likelihood estimation.

More generally, our methodology will allow us to adapt the noise schedule to the specific characteristics of an input image. For example, a set of photographs of the sky might benefit from a different schedule than images of human faces. When a diffusion process introduces noise into an image, it tends to effectively mask high frequency components first (e.g., by drowning the textures in white noise), while low-level frequencies (e.g., large shapes) disappear in noise last. Thus, images with varying amounts of high-level and low-level frequencies might benefit from varying noise schedules. A learned model might also discover improved noising strategies that we cannot anticipate.

## 3.2 A FORWARD DIFFUSION PROCESS WITH MULTIVARIATE ADAPTIVE NOISE

Next, our plan is to define a family of approximate posteriors $\mathcal{Q}$, as well as a family suitably matching reverse processes $\mathcal{P}$, such that the optimization problem $\max_{p \in \mathcal{P}, q \in \mathcal{Q}} \text{ELBO}(p, q)$ is tractable and does not suffer from the aforementioned invariance to the choice of $q$. This subsection focuses on defining $\mathcal{Q}$; the next sections will show how to parameterize and train a reverse model $p \in \mathcal{P}$.

**Notation.** Given two vectors $\mathbf{a}$ and $\mathbf{b}$, we use the notation $\mathbf{ab}$ to represent the Hadamard product (element-wise multiplication). Additionally, we denote element-wise division of $\mathbf{a}$ by $\mathbf{b}$ as $\mathbf{a} / \mathbf{b}$. We denote the mapping diag(.) that takes a vector as input and produces a diagonal matrix as output.

### 3.2.1 MULTIVARIATE GAUSSIAN NOISE SCHEDULE

Our proposed forward diffusion process progressively induces varying amounts of Gaussian noise across different areas of the image. We introduce two new components relative to previous work: multivariate noise scheduling and context-adaptive noise.

Intuitively, a multivariate noise schedule injects noise at different rates for each pixel of an input image. This enables adapting the diffusion process to spatial variations within the image. We will also see that this change is sufficient to make the ELBO no longer invariant in $q$.

Formally, our definition of a forward diffusion process with a multivariate noise schedule follows previous work (Kingma et al., 2021; Hoogeboom & Salimans, 2022) and defines $q$ via the marginal for each latent noise variable $\mathbf{x}_t$ for $t \in [0, 1]$, where the marginal is given by:

$$q(\mathbf{x}_t | \mathbf{x}_0) = \mathcal{N}(\mathbf{x}_t; \boldsymbol{\alpha}_t \mathbf{x}_0, \text{diag}(\boldsymbol{\sigma}_t^2)), \tag{2}$$

where $\mathbf{x}_t, \mathbf{x}_0 \in \mathbb{R}^d$, $\boldsymbol{\alpha}_t, \boldsymbol{\sigma}_t \in \mathbb{R}_+^d$ and $d$ is the dimensionality of the input data. The $\boldsymbol{\alpha}_t, \boldsymbol{\sigma}_t$ denote varying amounts of signal and associated with each component (i.e., each pixel) of $\mathbf{x}_0$ as a function of time $t(i)$. Similarly to Kingma et al. (2021), we may define the multivariate signal-to-noise ratio as $\boldsymbol{\nu}(t) = \boldsymbol{\alpha}_t^2 \boldsymbol{\sigma}_t^{-2}$ and we choose $\boldsymbol{\alpha}_t, \boldsymbol{\sigma}_t$ such that $\boldsymbol{\nu}(t)$ is monotonically decreasing in $t$ along all dimensions and is differentiable in $t \in [0, 1]$. Let $\boldsymbol{\alpha}_{t|s} = \boldsymbol{\alpha}_t / \boldsymbol{\alpha}_s$ and $\boldsymbol{\sigma}_{t|s}^2 = \boldsymbol{\sigma}_t^2 - \boldsymbol{\alpha}_{t|s}^2 / \boldsymbol{\sigma}_s^2$ with all operations applied elementwise. In Hoogeboom & Salimans (2022), show that these marginals induce transition kernels of the true reverse process between steps $s < t$ that are given by:

$$q(\mathbf{x}_s | \mathbf{x}_t, \mathbf{x}_0) = \mathcal{N}\left(\mathbf{x}_s; \boldsymbol{\mu}_q = \frac{\boldsymbol{\alpha}_{t|s} \boldsymbol{\sigma}_s^2}{\boldsymbol{\sigma}_t^2} \mathbf{x}_t + \frac{\boldsymbol{\sigma}_{t|s}^2 \boldsymbol{\alpha}_s}{\boldsymbol{\sigma}_t^2} \mathbf{x}_0, \; \boldsymbol{\Sigma}_q = \text{diag}\left(\frac{\boldsymbol{\sigma}_s^2 \boldsymbol{\sigma}_{t|s}^2}{\boldsymbol{\sigma}_t^2}\right)\right) \tag{3}$$

In Sec. 3.5, we argue that this class of diffusion process $\mathcal{Q}$ induces an ELBO that is not invariant to $q \in \mathcal{Q}$. The ELBO consists of a line integral along the diffusion trajectory specified by $\boldsymbol{\nu}(t)$. A line integrand is almost always path-dependent, unless its integral corresponds to a conservative force field, which is rarely the case for a diffusion process (Spinney & Ford, 2012). See Sec. 3.5 for details.

### 3.2.2 ADAPTIVE NOISE SCHEDULE CONDITIONED ON CONTEXT

Next, we extend the diffusion process to support context-adaptive noise. This enables injecting noise in a way that is dependent on the features of an image. Formally, we introduce a context variable

$\mathbf{c} \in \mathbb{R}^m$ which encapsulates high-level information regarding $\mathbf{x}_0$. Examples of $\mathbf{c}$ could be a class label, a vector of attributes (e.g., features characterizing a human face), or even the input $\mathbf{x}_0$ itself. We define the marginal of the latent $\mathbf{x}_t$ in the forward process as $q(\mathbf{x}_t|\mathbf{x}_0, \mathbf{c}) = \mathcal{N}(\mathbf{x}_t; \boldsymbol{\alpha}_t(\mathbf{c})\mathbf{x}_0, \boldsymbol{\sigma}_t^2(\mathbf{c}))$; the reverse process kernel can be similarly derived as Hoogeboom & Salimans (2022):

$$q(\mathbf{x}_s|\mathbf{x}_t, \mathbf{x}_0, \mathbf{c}) = \mathcal{N}\left(\boldsymbol{\mu}_q = \frac{\boldsymbol{\alpha}_{t|s}(\mathbf{c})\boldsymbol{\sigma}_s^2(\mathbf{c})}{\boldsymbol{\sigma}_t^2(\mathbf{c})}\mathbf{x}_t + \frac{\boldsymbol{\sigma}_{t|s}^2(\mathbf{c})\boldsymbol{\alpha}_s(\mathbf{c})}{\boldsymbol{\sigma}_t^2(\mathbf{c})}\mathbf{x}_0, \ \boldsymbol{\Sigma}_q = \text{diag}\left(\frac{\boldsymbol{\sigma}_s^2(\mathbf{c})\boldsymbol{\sigma}_{t|s}^2(\mathbf{c})}{\boldsymbol{\sigma}_t^2(\mathbf{c})}\right)\right)$$
(4)

where the diffusion parameters $\boldsymbol{\alpha}_t, \boldsymbol{\sigma}_t$ are now conditioned on $\mathbf{c}$ via a neural network.

Specifically, we parameterize the diffusion parameters $\boldsymbol{\alpha}_t(\mathbf{c}), \boldsymbol{\sigma}_t(\mathbf{c}), \boldsymbol{\nu}(t, \mathbf{c})$ as $\boldsymbol{\alpha}_t^2(\mathbf{c}) = \text{sigmoid}(-\boldsymbol{\gamma}_\phi(\mathbf{c}, t))$, $\boldsymbol{\sigma}_t^2(\mathbf{c}) = \text{sigmoid}(\boldsymbol{\gamma}_\phi(\mathbf{c}, t))$, and $\boldsymbol{\nu}(\mathbf{c}, t) = \exp(-\boldsymbol{\gamma}_\phi(\mathbf{c}, t))$. Here, $\boldsymbol{\gamma}_\phi(\mathbf{c}, t) : \mathbb{R}^m \times [0, 1] \to [\gamma_{\min}, \gamma_{\max}]^d$ is a neural network with the property that $\boldsymbol{\gamma}_\phi(\mathbf{c}, t)$ is monotonic in $t$. Following Kingma et al. (2021); Zheng et al. (2023), we set $\gamma_{\min} = -13.30, \gamma_{\max} = 5.0$.

We explore various parameterizations for $\boldsymbol{\gamma}_\phi(\mathbf{c}, t)$. These schedules are designed in a manner that guarantees $\boldsymbol{\gamma}_\phi(\mathbf{c}, 0) = \gamma_{\min}\mathbf{I}_n$ and $\boldsymbol{\gamma}_\phi(\mathbf{c}, 1) = \gamma_{\max}\mathbf{I}_n$., Below, we list these parameterizations. The polynomial parameterization is novel to our work and yields significant performance gains.

**Monotonic Neural Network.** (Kingma et al., 2021) We use the monotonic neural network $\gamma_{\text{vdm}}(t)$, proposed in VDM to express $\gamma$ as a function of $t$ such that $\gamma_{\text{vdm}}(t) : [0, 1] \to [\gamma_{\min}, \gamma_{\max}]^d$. Then we use FiLM conditioning (Perez et al., 2018) in the intermediate layers of this network via a neural network that maps $\mathbf{z}$. The activations of the FiLM layer are constrained to be positive.

**Sigmoid.** (Ours) We express $\boldsymbol{\gamma}_\phi(\mathbf{c}, t)$ as a sigmoid function in $t$ such that:
$\gamma(\mathbf{c}, t) = \gamma_{\min} + (\gamma_{\max} - \gamma_{\min}) \frac{\sigma(\mathbf{a}(\mathbf{c})t + \mathbf{b}(\mathbf{c})) - \sigma(\mathbf{b}(\mathbf{c}))}{\sigma(\mathbf{a}(\mathbf{c}) + \mathbf{b}(\mathbf{c})) - \sigma(\mathbf{b}(\mathbf{c}))}$ where $\sigma(\mathbf{c}) = 1/(1 + e^{-\mathbf{c}})$. Coefficients $\mathbf{a}, \mathbf{b}$ are parameterized by a neural network such that $\mathbf{a} : \mathbb{R}^m \to \mathbb{R}^{d+}, \mathbf{b} : \mathbb{R}^m \to \mathbb{R}^d$.

**Polynomial.** (Ours) We express $\gamma$ as a degree 5 polynomial in $t$. $\gamma(\mathbf{c}, t) = \gamma_{\min} + (\gamma_{\max} - \gamma_{\min}) \left(\frac{\mathbf{a}(\mathbf{c})t^5 + \mathbf{b}(\mathbf{c})t^4 + \mathbf{d}(\mathbf{c})t^3 + \mathbf{e}(\mathbf{c})t^2 + \mathbf{f}(\mathbf{c})t}{\mathbf{a}(\mathbf{c}) + \mathbf{b}(\mathbf{c}) + \mathbf{d}(\mathbf{c}) + \mathbf{e}(\mathbf{c}) + \mathbf{f}(\mathbf{c})}\right)$ In Suppl. D.2 we discuss the tricks we used to ensure the monotonicity of $\boldsymbol{\gamma}_\phi(\mathbf{z}, t)$ in $t$.

### 3.3 AUXILIARY-VARIABLE REVERSE DIFFUSION PROCESSES

In principle, we can fit a normal diffusion model in conjunction with our proposed forward diffusion process. However, variational inference suggests that the variational and the true posterior ought to have the same dependency structure: that is the only way for the KL divergence between these two distributions to be zero. Thus, we introduce a class of approximate reverse processes $\mathcal{P}$ that match the structure of $\mathcal{Q}$ and that are naturally suited to the joint optimization $\max_{p \in \mathcal{P}, q \in \mathcal{Q}} \text{ELBO}(p, q)$.

Formally, we define a diffusion model where the reverse diffusion process is conditioned on the context $\mathbf{c}$. Specifically, given any discretization of $t \in [0, 1]$ into $T$ time steps as in Sec. 2, we introduce a context-conditional diffusion model $p_\theta(\mathbf{x}_{0:1}|\mathbf{c})$ that factorizes as the Markov chain

$$p_\theta(\mathbf{x}_{0:1}|\mathbf{c}) = p_\theta(\mathbf{x}_1|\mathbf{c}) \prod_{i=1}^{T} p_\theta(\mathbf{x}_{s(i)}|\mathbf{x}_{t(i)}, \mathbf{c}).$$
(5)

Given that the true reverse process is a Gaussian as specified in Eq. 4, the ideal $p_\theta$ matches this parameterization (the proof mirrors that of regular diffusion models; Suppl. C), which yields

$$p_\theta(\mathbf{x}_s|\mathbf{c}, \mathbf{x}_t) = \mathcal{N}\left(\boldsymbol{\mu}_p = \frac{\boldsymbol{\alpha}_{t|s}(\mathbf{c})\boldsymbol{\sigma}_s^2(\mathbf{c})}{\boldsymbol{\sigma}_t^2(\mathbf{c})}\mathbf{x}_t + \frac{\boldsymbol{\sigma}_{t|s}^2(\mathbf{c})\boldsymbol{\alpha}_s(\mathbf{c})}{\boldsymbol{\sigma}_t^2(\mathbf{c})}\mathbf{x}_\theta(\mathbf{x}_0, t), \ \boldsymbol{\Sigma}_p = \text{diag}\left(\frac{\boldsymbol{\sigma}_s^2(\mathbf{c})\boldsymbol{\sigma}_{t|s}^2(\mathbf{c})}{\boldsymbol{\sigma}_t^2(\mathbf{c})}\right)\right),$$
(6)

where $\mathbf{x}_\theta : (\mathbf{x}_t, t)$, is a neural network that approximates $\mathbf{x}_0$. We parameterize the denoising model $\mathbf{x}_\theta$ in terms of a noise prediction model (Ho et al., 2020) where $\epsilon_\theta(\mathbf{x}_t, t)$ is the denoising model which is parameterized as $\epsilon_\theta(\mathbf{x}_t, t) = (\mathbf{x}_t - \boldsymbol{\alpha}_t(\mathbf{c})\mathbf{x}_\theta(\mathbf{x}_t; t, \mathbf{c}))/\boldsymbol{\sigma}_t(\mathbf{c})$ (see Suppl. D.1).

#### 3.3.1 CHALLENGES IN CONDITIONING ON CONTEXT

Note that the model $p_\theta(\mathbf{x}_{0:1}|\mathbf{c})$ implicitly assumes the availability of $\mathbf{c}$ at generation time. Sometimes, this context may be available, such as when we condition on a label. We may then fit a conditional

diffusion process with a standard diffusion objective $\mathbb{E}_{\mathbf{x}_0, c}[\text{ELBO}(\mathbf{x}_0, p_\theta(\mathbf{x}_{0:1}|\mathbf{c}), q_\phi(\mathbf{x}_{0:1}|\mathbf{c})]$, in which both the forward and the backward processes are conditioned on $\mathbf{c}$ (see Sec. 3.4).

When $\mathbf{c}$ is not known at generation time, we may fit a model $p_\theta$ that does not condition on $\mathbf{c}$. Unfortunately, this also forces us to define $p_\theta(\mathbf{x}_s|\mathbf{x}_t) = \mathcal{N}(\boldsymbol{\mu}_p(\mathbf{x}_t, t), \boldsymbol{\Sigma}_p(\mathbf{x}_t, t))$ where $\boldsymbol{\mu}_p(\mathbf{x}_t, t), \boldsymbol{\Sigma}_p(\mathbf{x}_t, t)$ is parameterized directly by a neural network. We can no longer use a noise parameterization $\epsilon_\theta(\mathbf{x}_t, t) = (\mathbf{x}_t - \boldsymbol{\alpha}_t(\mathbf{c})\mathbf{x}_\theta(\mathbf{x}_t; t, \mathbf{c}))/\boldsymbol{\sigma}_t(\mathbf{c})$ because it requires us to compute $\boldsymbol{\alpha}_t(\mathbf{c})$ and $\boldsymbol{\sigma}_t(\mathbf{c})$, which we do not know. Since noise parameterization plays a key role in the sample quality of diffusion models (Ho et al., 2020), this approach limits performance.

The other approach is to approximate $\mathbf{c}$ using a neural network, $\mathbf{c}_\theta(\mathbf{x}_t, t)$. This would allow us to write $p_\theta(\mathbf{x}_s|\mathbf{x}_t) = q_\phi(\mathbf{x}_s|\mathbf{x}_t, \mathbf{x}_0 = \mathbf{x}_\theta(\mathbf{x}_t, t), \mathbf{c} = \mathbf{c}_\theta(\mathbf{x}_t, t))$. Unfortunately, this introduces instability in the learning objective, which we observe both theoretically and empirically. Specifically, in Suppl. C we show that the learning objective diverges unless the following condition holds true: $\lim_{T \to \infty} T \frac{\boldsymbol{\sigma}_t^2(\mathbf{x}_0)\boldsymbol{\nu}_t(\mathbf{x}_0)\boldsymbol{\nu}_t'(\mathbf{x}_\theta)}{\boldsymbol{\sigma}_t^2(\mathbf{x}_\theta)\boldsymbol{\nu}_t(\mathbf{x}_\theta)\boldsymbol{\nu}_t'(\mathbf{x}_0)} \to \mathbf{I}_d$ pointwise across t. Experiments in Suppl. C.3 confirm this issue.

### 3.3.2 CONDITIONING NOISE ON AN AUXILIARY LATENT VARIABLE

Instead, we propose an alternative strategy for learning conditional forward and reverse processes $p, q$ that feature the same structure and hence support efficient noise parameterization. Our approach is based on the introduction of auxiliary variables (Wang et al., 2023), which lift the distribution $p_\theta$ into an augmented latent space.

Specifically, we define $\mathbf{z} \in \mathbb{R}^m$ as a low-dimensional auxiliary latent variable and define a lifted $p_\theta(\mathbf{x}, \mathbf{z}) = p_\theta(\mathbf{x}|\mathbf{z})p_\theta(\mathbf{z})$, where $p_\theta(\mathbf{x}|\mathbf{z})$ is the conditional diffusion model from Eq. 5 (with context $\mathbf{c}$ set to $\mathbf{z}$) and $p_\theta(\mathbf{z})$ is a simple prior (e.g., unit Gaussian or fully factored Bernoulli). The latents $\mathbf{z}$ can be interpreted as a high-level semantic representation of $\mathbf{x}$ that conditions both the forward and the reverse processes. Unlike $\mathbf{x}_{0:1}$, the $\mathbf{z}$ are not constrained to have a particular dimension and can be a low-dimensional vector of latent factors of variation. They can be either continuous or discrete.

We form a learning objective for the lifted $p_\theta$ by applying the ELBO twice to obtain:

$$\log p_\theta(\mathbf{x}_0) \geq \mathbb{E}_{q_\phi(\mathbf{z}|\mathbf{x}_0)}[\log p_\theta(\mathbf{x}_0|\mathbf{z})] - D_{\text{KL}}(q_\phi(\mathbf{z}|\mathbf{x}_0)\|p_\theta(\mathbf{z})) \tag{7}$$

$$\geq \mathbb{E}_{q_\phi(\mathbf{z}|\mathbf{x}_0)}[\text{ELBO}(p_\theta(\mathbf{x}_{0:1}|\mathbf{z}), q_\phi(\mathbf{x}_{0:1}|\mathbf{z}))] - D_{\text{KL}}(q_\phi(\mathbf{z}|\mathbf{x}_0)\|p_\theta(\mathbf{z})), \tag{8}$$

where $\text{ELBO}(p_\theta(\mathbf{x}_{0:1}|\mathbf{z}), q_\phi(\mathbf{x}_{0:1}|\mathbf{z}))$ denotes the variational lower bound of a diffusion model (defined in Eq. 1) with a forward process $q_\phi(\mathbf{x}_{0:1}|\mathbf{z})$ (defined in Eq. 4 and Sec. 3.2.2) and and an approximate reverse process $p_\theta(\mathbf{x}_{0:1}|\mathbf{z})$ (defined in Eq. 5), both conditioned on $\mathbf{z}$. The distribution $q_\phi(\mathbf{z}|\mathbf{x}_0)$ is an approximate posterior for $\mathbf{z}$ parameterized by a neural network with parameters $\phi$.

Crucially, note that in the learning objective (8), the context, which in this case is $\mathbf{z}$, is available at training time in both the forward and reverse processes. At generation time, we can still obtain a valid context vector by sampling an auxiliary latent from $p_\theta(\mathbf{z})$. Thus, this approach addresses the aforementioned challenges and enables us to use the noise parameterization in Eq. 6.

## 3.4 VARIATIONAL LOWER BOUND

Next, we derive a precise formula for the learning objective (8) of the auxiliary-variable diffusion model. Using the objective of a diffusion model in (1) we can write (8) as a sum of four terms

$$\log p_\theta(\mathbf{x}_0) \geq \mathbb{E}_{q_\phi}[\mathcal{L}_{\text{reconstr}} + \mathcal{L}_{\text{diffusion}} + \mathcal{L}_{\text{prior}} + \mathcal{L}_{\text{latent}}], \tag{9}$$

where $\mathcal{L}_{\text{reconstr}} = \log p_\theta(\mathbf{x}_0|\mathbf{z}, \mathbf{x}_1)$ is the reconstruction error, $\mathcal{L}_{\text{prior}} = -D_{\text{KL}}[q_\phi(\mathbf{x}_1|\mathbf{x}_0, \mathbf{z})\|p_\theta(\mathbf{x}_1)]$ is the diffusion prior term, $\mathcal{L}_{\text{latent}} = -D_{\text{KL}}[q_\phi(\mathbf{z}|\mathbf{x}_0)\|p_\theta(\mathbf{z})]$ is the latent prior term, and $\mathcal{L}_{\text{diffusion}}$ is the diffusion loss term, which we examine below. The full derivation is in Suppl. D.4.

The reconstruction loss, $\mathcal{L}_{\text{recons}}$, and the prior loss, $\mathcal{L}_{\text{prior}}$, are stochastically and differentiablly estimated using standard techniques; see Kingma & Welling (2013). The diffusion loss $\mathcal{L}_{\text{diffusion}}$, and the latent loss $\mathcal{L}_{\text{latent}}$ are computed in the manner examined below.

### 3.4.1 DIFFUSION LOSS

**Discrete-Time Diffusion.** We start by defining $p_\theta$ in discrete time, and as in Sec. 2, we let $T > 0$ be the number of total time steps and define $t(i) = i/T$ and $s(i) = (i-1)/T$ as indexing variables

over the time steps. We also use $\mathbf{x}_{0:1}$ to denote the subset of variables associated with these timesteps. Starting with the expression in Eq. 1 and following the steps in Suppl. D, we can write $\mathcal{L}_{\text{diffusion}}$ as:

$$\mathcal{L}_{\text{diffusion}} = -\sum_{i=2}^{T} \mathrm{D}_{\text{KL}}[q_\phi(\mathbf{x}_{s(i)}|\mathbf{x}_{t(i)}, \mathbf{x}_0, \mathbf{z})\|p_\theta(\mathbf{x}_{s(i)}|\mathbf{x}_{t(i)}, \mathbf{z})]$$

$$= \frac{1}{2}\sum_{i=2}^{T} \left[(\epsilon_t - \epsilon_\theta(\mathbf{x}_t, \mathbf{z}, t(i)))^\top \text{diag}\left(\boldsymbol{\gamma}(\mathbf{z}, s(i)) - \boldsymbol{\gamma}(\mathbf{z}, t(i))\right)(\epsilon_t - \epsilon_\theta(\mathbf{x}_t, \mathbf{z}, t(i)))\right] \quad (10)$$

**Continuous-Time Diffusion.**   We can also consider the limit of the above objective as we take an infinitesimally small partition of $t \in [0, 1]$, which corresponds to the limit when $T \to \infty$. In Suppl. D we show that taking this limit of Eq. 10 yields the continuous-time diffusion loss:

$$\mathcal{L}_{\text{diffusion}} = -\frac{1}{2}\mathbb{E}_{t\sim[0,1]}\left[(\epsilon_t - \epsilon_\theta(\mathbf{x}_t, \mathbf{z}, t))^\top \text{diag}\left(\nabla_t\boldsymbol{\gamma}(\mathbf{z}, t)\right)(\epsilon_t - \epsilon_\theta(\mathbf{x}_t, \mathbf{z}, t))\right] \quad (11)$$

where $\nabla_t\boldsymbol{\gamma}(\mathbf{z}, t) \in \mathbb{R}^d$ denotes the Jacobian of $\boldsymbol{\gamma}(\mathbf{z}, t)$ with respect to the scalar $t$. We observe that the limit of $T \to \infty$ yields improved performance, matching the existing theoretical argument by Kingma et al. (2021).

### 3.4.2   AUXILIARY LATENT LOSS

We try two different kinds of priors for $p_\theta(\mathbf{z})$: discrete ($\mathbf{z} \in \{0, 1\}^m$) and continuous ($\mathbf{z} \in \mathbb{R}^m$).

**Continuous Auxiliary Latents.**   In the case where $\mathbf{z}$ is continuous, we select $p_\theta(\mathbf{z})$ as $\mathcal{N}(\mathbf{0}, \mathbf{I}_m)$. This leads to the following KL loss term:
$\mathrm{D}_{\text{KL}}(q_\phi(\mathbf{z}|\mathbf{x}_0)\|p_\theta(\mathbf{z})) = \frac{1}{2}(\boldsymbol{\mu}^\top(\mathbf{x}_0)\boldsymbol{\mu}(\mathbf{x}_0)) + \text{tr}(\boldsymbol{\Sigma}^2(\mathbf{x}_0) - \mathbf{I}_m) - \log|\boldsymbol{\Sigma}^2(\mathbf{x}_0)|).$

**Discrete Auxiliary Latents.**   In the case where $\mathbf{z}$ is discrete, we select $p_\theta(\mathbf{z})$ as a uniform distribution. Let $\mathbf{z} \in \{0, 1\}^m$ be a $k$-hot vector sampled from a discrete Exponential Family distribution $p_\theta(\mathbf{z}; \theta)$ with logits $\theta$. Niepert et al. (2021) show that $\mathbf{z} \sim p_\theta(\mathbf{z}; \theta)$ is equivalent to $\mathbf{z} = \arg\max_{y\in Y}\langle\theta + \epsilon_g, y\rangle$ where $\epsilon_g$ denotes the sum of gamma distribution Suppl. D.3, $Y$ denotes the set of all $k$-hot vectors of some fixed length $m$. For $k > 1$, To differentiate through the $\arg\max$ we use a relaxed estimator, Identity, as proposed by Sahoo et al. (2023). This leads to the following KL loss term: $\mathrm{D}_{\text{KL}}(q_\phi(\mathbf{z}|\mathbf{x}_0)\|p_\theta(\mathbf{z})) = -\sum_{i=1}^{m} q_\phi(\mathbf{z}|\mathbf{x}_0)_i \log(m q_\phi(\mathbf{z}|\mathbf{x}_0)_i).$

### 3.5   THE VARIATIONAL LOWER BOUND AS A LINE INTEGRAL OVER THE NOISE SCHEDULE

Having defined our loss, we now return to the question of whether it is invariant to the choice of diffusion process. Notice that we may rewrite Eq. 11 in the following vectorized form:

$$\mathcal{L}_{\text{diffusion}} = -\frac{1}{2}\int_0^1 (\mathbf{x}_0 - \mathbf{x}_\theta(\mathbf{x}_t, \mathbf{z}, t))^2 \cdot \nabla_t\boldsymbol{\nu}(\mathbf{z}, t)dt \quad (12)$$

where the square is applied elementwise. We seek to rewrite (12) as a line integral $\int_a^b \mathbf{f}(\mathbf{r}(t)) \cdot \frac{d}{dt}\mathbf{r}(t)dt$ for some vector field $\mathbf{f}$ and trajectory $\mathbf{r}(t)$. Recall that $\boldsymbol{\nu}(\mathbf{z}, t)$ is monotonically decreasing in each coordinate as a function of $t$; hence, it is invertible on its image, and we can write $t = \boldsymbol{\nu}_{\mathbf{z}}^{-1}(\boldsymbol{\nu}(\mathbf{z}, t))$ for some $\boldsymbol{\nu}_z^{-1}$. Let $\bar{\mathbf{x}}_\theta(\mathbf{x}_{\boldsymbol{\nu}(\mathbf{z},t)}, \mathbf{z}, \boldsymbol{\nu}(\mathbf{z}, t)) = \mathbf{x}_\theta(\mathbf{x}_{\boldsymbol{\nu}_z^{-1}(\boldsymbol{\nu}(\mathbf{z},t))}, \mathbf{z}, \boldsymbol{\nu}_z^{-1}(\boldsymbol{\nu}(\mathbf{z}, t)))$ and note that for all $t$, we can write $\mathbf{x}_t$ as $\mathbf{x}_{\boldsymbol{\nu}(\mathbf{z},t)}$; see Eq. 26, and have $\bar{\mathbf{x}}_\theta(\mathbf{x}_{\boldsymbol{\nu}(\mathbf{z},t)}, \mathbf{z}, \boldsymbol{\nu}(\mathbf{z}, t)) = \mathbf{x}_\theta(\mathbf{x}_t, \mathbf{z}, t)$. We can then write the integral in (12) as $\int_0^1 (\mathbf{x}_0 - \bar{\mathbf{x}}_\theta(\mathbf{x}_{\boldsymbol{\nu}(\mathbf{z},t)}, \mathbf{z}, \boldsymbol{\nu}(\mathbf{z}, t)))^2 \cdot \frac{d}{dt}\boldsymbol{\nu}(\mathbf{z}, t)\rangle dt$, which is a line integral with $\mathbf{f} = (\mathbf{x}_0 - \bar{\mathbf{x}}_\theta(\mathbf{x}_{\boldsymbol{\nu}(\mathbf{z},t)}, \mathbf{z}, \boldsymbol{\nu}(\mathbf{z}, t)))^2$ and $\mathbf{r}(t) = \boldsymbol{\nu}(\mathbf{z}, t)$.

Thus the diffusion loss, $\mathcal{L}_{\text{diffusion}}$, can be interpreted as a measure of work done along the trajectory $\boldsymbol{\nu}(\mathbf{z}, t)$ in the presence of a vector field $\mathbf{f}$. Different "trajectories" yield different results for most integrands, unless $\mathbf{f}$ is conservative. Although we do not guarantee that no model or dataset will yield a conservative $\mathbf{f}$, we observe empirically that across our experiments, swapping out different multivariate $\boldsymbol{\nu}$ yields different values of the ELBO. In D.6, we show that variational diffusion models can be viewed as following only linear trajectories $\boldsymbol{\nu}(t)$, hence their objective is invariant to the noise schedule. Our method learns a multivariate $\boldsymbol{\nu}$ that yields paths corresponding to a better ELBO.

Table 1: Likelihood in bits per dimension (BPD) on the test set of CIFAR-10 and ImageNet. Results with "/" means they are not reported in the original papers. Model types are autoregressive (AR), normalizing flows (Flow), variational autoencoders (VAE), or diffusion models (Diff).

| Model | Type | CIFAR-10 ($\downarrow$) | ImageNet ($\downarrow$) |
|---|---|---|---|
| PixelCNN (Van den Oord et al., 2016) | AR | 3.03 | 3.83 |
| PixelCNN++ (Salimans et al., 2017) | AR | 2.92 | / |
| Glow (Kingma & Dhariwal, 2018) | Flow | / | 4.09 |
| Image Transformer (Parmar et al., 2018) | AR | 2.90 | 3.77 |
| DDPM (Ho et al., 2020) | Diff | 3.69 | / |
| Score SDE (Song et al., 2020) | Diff | 2.99 | / |
| Improved DDPM (Nichol & Dhariwal, 2021) | Diff | 2.94 | / |
| VDM (Kingma et al., 2021) | Diff | 2.65 | 3.72 |
| Flow Matching (Lipman et al., 2022) | Flow | 2.99 | / |
| i-DODE* (Zheng et al., 2023) | Diff | 2.61 | / |
| i-DODE* (Zheng et al., 2023) | Flow | 2.56 | 3.69 |
| MuLAN (**Ours**) | Diff | 2.60 | 3.71 |
| MuLAN (**Ours**) | Flow | **2.55** | **3.67** |

## 4 EXPERIMENTS

This section reports experimental results on the CIFAR-10 (Krizhevsky et al., 2009) and downsampled ImageNet (Van Den Oord et al., 2016) datasets.

**Setup** We implement our methods based on the open-source codebase of Kingma et al. (2021) using JAX (Bradbury et al., 2018), and use the exact same hyperparameter settings and network architecture for the denoising model. Our network that models $q_\phi(\mathbf{z}|\mathbf{x})$ is a sequence of 4 Resnet blocks with the number of channels set to 128 which is much smaller than the denoising network which has 64 such blocks. Like VDM, we train our models for 10M steps. We chose to employ a discrete prior for the auxiliary latent space rather than a Gaussian prior due to training instability issues that frequently led to NaNs. In all our experiments, we set the parameters for the discrete latent distribution as $m = 32$ and $k = 8$.

### 4.1 LIKELIHOOD ESTIMATION.

In Table 1, we present the likelihood estimation results for MuLAN, and other recent methods on CIFAR-10 and ImageNet. We apply MuLAN on top of the VDM model (Kingma et al., 2021), endowing it with a learned multivariate noising schedule conditioned on auxiliary latent variables. We find that these new components result in a significant improvement in BPD over a vanilla VDM.

Overall, our method outperforms all existing published works except for Zheng et al. (2023). Their approach formulates the diffusion models as an ODE and employs a new parameterization of $p_\theta(\mathbf{x}_s|\mathbf{x}_t)$ as well as an importance sampled gradient estimator. These improvements to the model are orthogonal to our work on the forward process, and we expect that combining them can further improve performance.

Additionally, our method also improves training speed. Specifically, MuLAN achieves an equivalent BPD score to a vanilla VDM on CIFAR-10 after 3M training steps, whereas VDM was trained to 10M steps to reach the same score. Likewise, on ImageNet, MuLAN achieves a matching BPD score with VDM after 1M training steps, while VDM was trained for 2M steps.

### 4.2 ABLATION ANALYSIS

Due to the expensive cost of training, we only conduct ablation studies on CIFAR-10 with a reduced batch size of 64 and train the model for 2.5M training steps. In Fig. 1 we ablate each component of MuLAN: when we remove the conditioning on an auxiliary latent space from MuLAN such that we have a multivariate noise schedule that is solely conditioned on time $t$, our performance becomes comparable to that of VDM, on which our model is based. Modifying our method to have a scalar

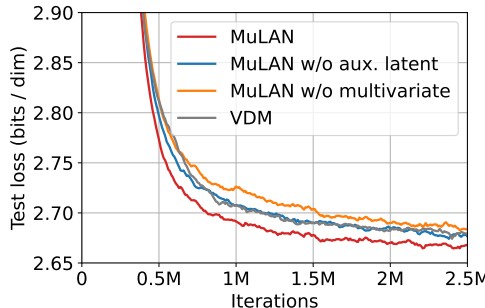
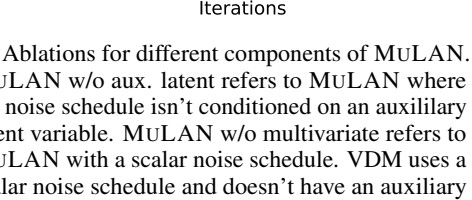
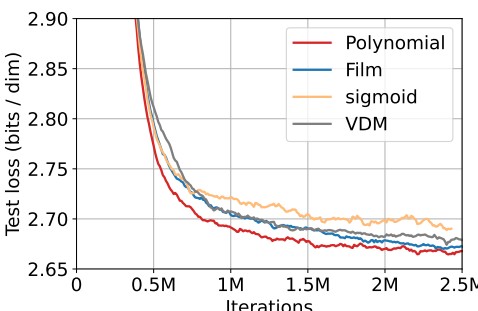

(a) Ablations for different components of MULAN. MULAN w/o aux. latent refers to MULAN where the noise schedule isn't conditioned on an auxiliary latent variable. MULAN w/o multivariate refers to MULAN with a scalar noise schedule. VDM uses a scalar noise schedule and doesn't have an auxiliary latent space. We see that MULAN performs the best.

(b) Ablations for noise schedules with different functional parameterizations: sigmoid, polynomial, and a monotonic neural network. We observe that the polynomial parameterization performs the best.

Figure 1: Ablation studies were conducted on on CIFAR-10 with a reduced batch size and fewer training steps.

noise schedule conditioned on the auxiliary latent variable $\mathbf{z}$ leads to slightly lower performance than VDM in the initial training stages. However, it gradually converges toward VDM.

**Loss curves for different noise schedules.** We investigate different parameterizations of the noise schedule in Fig. 1. Among polynomial, sigmoid, and monotonic neural network, we find that the polynomial parameterization yields the best performance. The polynomial noise schedule is a novel component introduced in our work.

**Replacing the noise schedules in a trained denoising model.** We also wish to confirm experimentally our claim that the learning objective is not invariant to our choice of multivariate noise schedule. To investigate this, we replace the noise schedule in the trained denoising model with two alternatives: MULAN with scalar noise schedule, and a linear noise schedule: $\boldsymbol{\gamma}_\phi(\mathbf{z}, t) = (\boldsymbol{\gamma}_{\min} + (\boldsymbol{\gamma}_{\max} - \boldsymbol{\gamma}_{\min})t)\mathbf{1_d}$; see (Kingma et al., 2021). For both the noise schedules the likelihood worsens to the same value as that of the VDM: 2.65. This experimental result strongly supports our theory that all scalar noise schedules are equivalent, as they compute the likelihood along the same diffusion trajectory. It also underscores that it's not the multivariate nature or the auxiliary latent space individually, but the combination of both, that makes MULAN effective.

**Examining the noise schedule.** Since the noise schedule, $\boldsymbol{\gamma}_\phi(\mathbf{z}, t)$ is multivariate, we expect to learn different noise schedules for different input dimensions and different inputs $\mathbf{z} \sim p_\theta(\mathbf{z})$. In Fig. 2, we take our best model trained on CIFAR-10 and we visualize the variance of the noise schedule at each point in time for different pixels, where the variance is taken over 128 samples $\mathbf{z} \sim p_\theta(\mathbf{z})$. We note increased variation in the early portions of the noise schedule. However, on an absolute scale, the variance of this noise is smaller than we expected. We also tried to visualize noise schedules across different dataset images and across different areas of the same image. We also generated synthetic datasets in which each datapoint contained only high frequencies or only low frequencies. Surprisingly, none of these experiments revealed human-interpretable patterns

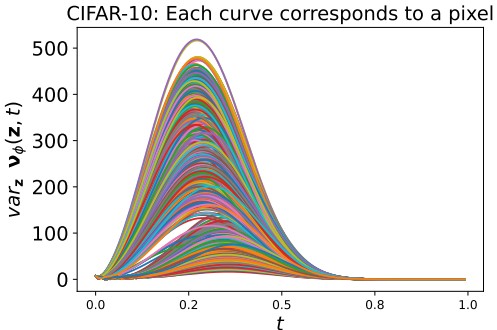

Figure 2: Noise schedule visualizations for MULAN on CIFAR-10. In this figure, we plot the variance of $\boldsymbol{\nu}_\phi(\mathbf{z}, t)$ across different $\mathbf{z} \sim p_\theta(\mathbf{z})$ where each curve represents the SNR corresponding to an input dimension.

in the learned schedule, although we did observe
clear differences in likelihood estimation. We hy-
pothesize that other architectures and other forms of conditioning may reveal interpretable patterns of
variation; however, we leave this exploration to future work.

## 5 RELATED WORK AND DISCUSSION

Diffusion models have emerged in recent years as powerful tools for modeling complex distributions
(Sohl-Dickstein et al., 2015; Song et al., 2020; Ho et al., 2020). The noise schedule, determining
the amount and type of noise added at each step, plays a critical role in diffusion models. Chen
(2023) emphasize that different noise schedules can significantly impact image quality in generated
outputs, though they do not learn these schedules but evaluate models under various handcrafted noise
schedules. Kingma et al. (2021) showed that the likelihood of a diffusion model remains invariant
to the noise schedule with a scalar noise schedule. We also take inspiration from the principles of
physics (Spinney & Ford, 2012) and show that the ELBO is no longer invariant to multivariate noise
schedules.

Recent works, including Hoogeboom & Salimans (2022); Rissanen et al. (2022); Pearl et al. (2023),
have explored per-pixel noise schedules, yet none have delved into learning or conditioning the
noise schedule on the input data itself. Yang & Mandt (2023); Wang et al. (2023) have explored
diffusion models with an auxiliary latent space, where the denoising network is conditioned on a
latent distribution. In contrast, our model conditions the noise schedule on the latent distribution,
promoting the clustering of inputs that would benefit from similar noise processes.

Song et al. (2020) show that the continuous formulation of the diffusion process Kingma et al.
(2021) can be modeled using a stochastic differential equation (SDE) and the marginals of the
corresponding reverse process for this SDE can be modeled using an equivalent ODE called the
diffusion ODE.Notably, diffusion ODEs represent special formulations of neural ODEs, akin to
continuous normalizing flows (Chen et al., 2018; Si et al., 2022; 2023). Some train diffusion ODEs
using the exact likelihood evaluation formula of ODE (Grathwohl et al., 2018). Given the expensive
ODE simulations, another approach involves training neural ODEs by matching their trajectories to
a predefined path, such as the diffusion process (Song et al., 2020; Lu et al., 2022; Lipman et al.,
2022; Liu et al., 2022). Zheng et al. (2023) train a diffusion model with a flow-matching objective
and importance sampling, differing from the approach used in VDM (Kingma et al., 2021).

## 6 CONCLUSION

In this study, we introduce MuLAN, a context-adaptive noise process that applies Gaussian noise
at varying rates across the input data. We present theoretical arguments challenging the prevailing
notion that the likelihood of diffusion models remains independent of the noise schedules. We
contend that this independence only holds true for univariate schedules, and in the case of multivariate
schedules like MuLAN, different diffusion trajectories yield distinct likelihood estimates. Our
evaluation of MuLAN spans multiple image datasets, where it outperforms state-of-the-art generative
diffusion models. We anticipate that our approach will stimulate further research into the design of
noise schedules, not only for improving likelihood estimation but also for enhancing image quality
generation (Parmar et al., 2018; Song et al., 2020); a stronger fit to the data distribution also holds
promise to improve downstream applications of generative modeling, e.g., decision making or causal
effect estimation (Nguyen & Grover, 2022; Deshpande et al., 2022; Deshpande & Kuleshov, 2023;
Rastogi et al., 2023). Broadly speaking, MuLAN represents a promising avenue for advancing
generative modeling.

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

# A  STANDARD DIFFUSION MODELS

We have a Gaussian diffusion process that begins with the data $\mathbf{x}_0$, and defines a sequence of increasingly noisy versions of $\mathbf{x}_0$ which we call the latent variables $\mathbf{x}_t$, where $t$ runs from $t = 0$ (least noisy) to $t = 1$ (most noisy). Given, $T$, we discretize time uniformly into $T$ timesteps each with a width $1/T$. We define $t(i) = i/T$ and $s(i) = (i-1)/T$.

## A.1  FORWARD PROCESS

$$q(\mathbf{x}_t|\mathbf{x}_s) = \mathcal{N}(\alpha_{t|s}\mathbf{x}_0, \sigma_{t|s}^2\mathbf{I}_n) \tag{13}$$

where

$$\alpha_{t|s} = \frac{\alpha_t}{\alpha_s} \tag{14}$$

$$\sigma_{t|s}^2 = \sigma_t^2 - \frac{\alpha_{t|s}^2}{\sigma_s^2} \tag{15}$$

## A.2  REVERSE PROCESS

Kingma et al. (2021) show that the distribution $q(\mathbf{x}_s|\mathbf{x}_t, \mathbf{x}_0)$ is also gaussian,

$$q(\mathbf{x}_s|\mathbf{x}_t, \mathbf{x}_0) = \mathcal{N}\left(\boldsymbol{\mu}_q = \frac{\alpha_{t|s}\sigma_s^2}{\sigma_t^2}\mathbf{x}_t + \frac{\sigma_{t|s}^2\alpha_s}{\sigma_t^2}x_0, \ \boldsymbol{\Sigma}_q = \frac{\sigma_s^2\sigma_{t|s}^2}{\sigma_t^2}\mathbf{I}_n\right) \tag{16}$$

Since during the reverse process, we don't have access to $\mathbf{x}_0$, we approximate it using a neural network $\mathbf{x}_\theta(\mathbf{x}_t, t)$ with parameters $\theta$. Thus,

$$p_\theta(\mathbf{x}_s|\mathbf{x}_t) = \mathcal{N}\left(\boldsymbol{\mu}_p = \frac{\alpha_{t|s}\sigma_s^2}{\sigma_t^2}\mathbf{x}_t + \frac{\sigma_{t|s}^2\alpha_s}{\sigma_t^2}\mathbf{x}_\theta(\mathbf{x}_t, t), \ \boldsymbol{\Sigma}_p = \frac{\sigma_s^2\sigma_{t|s}^2}{\sigma_t^2}\mathbf{I}_n\right) \tag{17}$$

## A.3  VARIATIONAL LOWER BOUND

This corruption process $q$ is the following markov-chain as $q(\mathbf{x}_{0:1}) = q(\mathbf{x}_0)\left(\prod_{i=1}^{T} q(\mathbf{x}_{t(i)}|\mathbf{x}_{s(i)})\right)$. In the reverse Rrocess, or the denoising process, $p_\theta$, a neural network (with parameters $\theta$) is used to denoise the noising process $q$. The reverse Rrocess factorizes as: $p_\theta(\mathbf{x}_{0:1}) = p_\theta(\mathbf{x}_1)\prod_{i=1}^{T} p_\theta(\mathbf{x}_{s(i)}|\mathbf{x}_{t(i)})$. Let $\mathbf{x}_\theta(\mathbf{x}_t, t)$ be the reconstructed input by a neural network from $\mathbf{x}_t$. Similar to Sohl-Dickstein et al. (2015); Kingma et al. (2021) we decompose the negative lower bound (VLB) as:

$$
\begin{aligned}
-\log p_\theta(\mathbf{x}_0) &\leq \mathbb{E}_{q_\phi}\left[-\log \frac{p_\theta(\mathbf{x}_{t(0):t(T)})}{q_\phi(\mathbf{x}_{t(1):t(T)}|\mathbf{x}_0)}\right] \\
&= \mathbb{E}_{\mathbf{x}_{t(1)}\sim q(\mathbf{x}_{t(1)}|\mathbf{x}_0)}[-\log p_\theta(\mathbf{x}_0|\mathbf{x}_{t(1)})] \\
&\quad + \sum_{i=2}^{T} \mathbb{E}_{\mathbf{x}_{t(i)}|\mathbf{x}_0}\mathrm{D}_{\mathrm{KL}}[p_\theta(\mathbf{x}_{s(i)}|\mathbf{x}_{t(i)})\|q_\phi(\mathbf{x}_{s(i)}|\mathbf{x}_{t(i)}, \mathbf{x}_0)] \\
&\quad + \mathrm{D}_{\mathrm{KL}}[p_\theta(\mathbf{x}_1)\|q_\phi(\mathbf{x}_1|\mathbf{x}_0)] \\
&= \underbrace{\mathbb{E}_{\mathbf{x}_{t(1)}\sim q(\mathbf{x}_{t(1)}|\mathbf{x}_0)}[-\log p_\theta(\mathbf{x}_0|\mathbf{x}_{t(1)})]}_{\mathcal{L}_{\mathrm{recons}}} \\
&\quad + \underbrace{\frac{T}{2}\mathbb{E}_{\epsilon\sim\mathcal{N}(0,\mathbf{I}_n),i\sim U\{2,T\}}\mathrm{D}_{\mathrm{KL}}[p_\theta(\mathbf{x}_{s(i)}|\mathbf{x}_{t(i)})\|q_\phi(\mathbf{x}_{s(i)}|\mathbf{x}_{t(i)}, \mathbf{x}_0)]}_{\mathcal{L}_{\mathrm{diffusion}}} \\
&\quad + \underbrace{\mathrm{D}_{\mathrm{KL}}[p_\theta(\mathbf{x}_1)\|q_\phi(\mathbf{x}_1|\mathbf{x}_0)]}_{\mathcal{L}_{\mathrm{prior}}}
\end{aligned} \tag{18}
$$

The prior loss, $\mathcal{L}_{\text{prior}}$, and reconstruction loss, $\mathcal{L}_{\text{recons}}$, can be (stochastically and differentiably) estimated using standard techniques; see Kingma & Welling (2013). The diffusion loss, $\mathcal{L}_{\text{diffusion}}$, varies with the formulation of the noise schedule. We provide an exact formulation for it in the subsequent sections.

### A.4 Diffusion Loss

For brevity, we use the notation $s$ for $s(i)$ and $t$ for $t(i)$. From Eq. 27 and Eq. 28 we get the following expression for $q(\mathbf{x}_s|\mathbf{x}_t, \mathbf{x}_0)$:

$$D_{\text{KL}}(q(\mathbf{x}_s|\mathbf{x}_t, \mathbf{x}_0)\|p_\theta(\mathbf{x}_s|\mathbf{x}_t))$$

$$= \frac{1}{2}\left((\boldsymbol{\mu}_q - \boldsymbol{\mu}_p)^\top \boldsymbol{\Sigma}_\theta^{-1}(\boldsymbol{\mu}_q - \boldsymbol{\mu}_p) + \text{tr}\left(\boldsymbol{\Sigma}_q \boldsymbol{\Sigma}_p^{-1} - \mathbf{I}_n\right) - \log\frac{|\boldsymbol{\Sigma}_q|}{|\boldsymbol{\Sigma}_p|}\right)$$

$$= \frac{1}{2}(\boldsymbol{\mu}_q - \boldsymbol{\mu}_p)^\top \Sigma_\theta^{-1}(\boldsymbol{\mu}_q - \boldsymbol{\mu}_p)$$

Substituting $\boldsymbol{\mu}_q, \boldsymbol{\Sigma}_q, \boldsymbol{\mu}_p, \boldsymbol{\Sigma}_p$ from equation 17 and equation 16; for the exact derivation see Kingma et al. (2021)

$$= \frac{1}{2}\left(\nu(s) - \nu(t)\right)\|(\mathbf{x}_0 - \mathbf{x}_\theta(\mathbf{x}_t, t))\|_2^2 \tag{19}$$

Thus $\mathcal{L}_{\text{diffusion}}$ is given by

$$\mathcal{L}_{\text{diffusion}}$$

$$= \lim_{T\to\infty}\frac{T}{2}\mathbb{E}_{\epsilon\sim\mathcal{N}(0,\mathbf{I}_n),i\sim U\{2,T\}}D_{\text{KL}}[p_\theta(\mathbf{x}_{s(i)}|\mathbf{x}_{t(i)})\|q_\phi(\mathbf{x}_{s(i)}|\mathbf{x}_{t(i)}, \mathbf{x}_0)]$$

$$= \lim_{T\to\infty}\frac{1}{2}\sum_{i=2}^{T}\mathbb{E}_{\epsilon\sim\mathcal{N}(0,\mathbf{I}_n)}\left(\nu(s) - \nu(t)\right)\|\mathbf{x}_0 - \mathbf{x}_\theta(\mathbf{x}_t, t)\|_2^2$$

$$= \frac{1}{2}\mathbb{E}_{\epsilon\sim\mathcal{N}(0,\mathbf{I}_n)}\left[\lim_{T\to\infty}\sum_{i=2}^{T}\left(\nu(s) - \nu(t)\right)\|\mathbf{x}_0 - \mathbf{x}_\theta(\mathbf{x}_t, t)\|_2^2\right]$$

$$= \frac{1}{2}\mathbb{E}_{\epsilon\sim\mathcal{N}(0,\mathbf{I}_n)}\left[\lim_{T\to\infty}\sum_{i=2}^{T}T\left(\nu(s) - \nu(t)\right)\|\mathbf{x}_0 - \mathbf{x}_\theta(\mathbf{x}_t, t)\|_2^2\frac{1}{T}\right]$$

Substituting $\lim_{T\to\infty}T(\nu(s) - \nu(t)) = \dfrac{d}{dt}\nu(t) \equiv \nu'(t)$; see Kingma et al. (2021)

$$= \frac{1}{2}\mathbb{E}_{\epsilon\sim\mathcal{N}(0,\mathbf{I}_n)}\left[\int_0^1 \nu'(t)\|\mathbf{x}_0 - \mathbf{x}_\theta(\mathbf{x}_t, t)\|_2^2\right]dt \tag{20}$$

In practice instead of computing the integral is computed by MC sampling.

$$= -\frac{1}{2}\mathbb{E}_{\epsilon\sim\mathcal{N}(0,\mathbf{I}_n),t\sim U[0,1]}\left[\nu'(t)\|\mathbf{x}_0 - \mathbf{x}_\theta(\mathbf{x}_t, t)\|_2^2\right] \tag{21}$$

## B Multivariate noise schedule

For a multivariate noise schedule we have $\boldsymbol{\alpha}_t, \boldsymbol{\sigma}_t \in \mathbb{R}^{n\times n}$ where $t \in [0, 1]$. $\boldsymbol{\alpha}_t, \boldsymbol{\sigma}_t$ are diagonal matrices. The timesteps $s, t$ satisfy $0 \le s < t \le 1$. Furthermore, we use the following notations where arithmetic division represents element wise division between 2 diagonal matrices:

$$\boldsymbol{\alpha}_{t|s} = \frac{\boldsymbol{\alpha}_t}{\boldsymbol{\alpha}_s} \tag{22}$$

$$\boldsymbol{\sigma}_{t|s}^2 = \boldsymbol{\sigma}_t^2 - \frac{\boldsymbol{\alpha}_{t|s}^2}{\boldsymbol{\sigma}_s^2} \tag{23}$$

## B.1 FORWARD PROCESS

$$q(\mathbf{x}_t|\mathbf{x}_s) = \mathcal{N}\left(\boldsymbol{\alpha}_{t|s}\mathbf{x}_s, \boldsymbol{\sigma}_{t|s}^2\right) \tag{24}$$

**Change of variables.** We can write $\mathbf{x}_t$ explicitly in terms of the signal-to-noise ratio, $\boldsymbol{\nu}(t)$, and input $\mathbf{x}_0$ in the following manner:

$$\boldsymbol{\nu}_t = \frac{\boldsymbol{\alpha}_t^2}{\boldsymbol{\sigma}_t^2}$$

We know $\alpha_t^2 = 1 - \sigma_t^2$ for Variance Preserving process; see Sec. 2.

$$\implies \frac{1 - \boldsymbol{\sigma}_t^2}{\boldsymbol{\sigma}_t^2} = \boldsymbol{\nu}_t$$

$$\implies \boldsymbol{\sigma}_t^2 = \frac{1}{1 + \boldsymbol{\nu}_t} \quad \text{and} \quad \boldsymbol{\alpha}_t^2 = \frac{\boldsymbol{\nu}_t}{1 + \boldsymbol{\nu}_t} \tag{25}$$

Thus, we write $\mathbf{x}_t$ in terms of the signal-to-noise ratio in the following manner:

$$\begin{aligned} \mathbf{x}_{\boldsymbol{\nu}(t)} &= \boldsymbol{\alpha}_t\mathbf{x}_0 + \boldsymbol{\sigma}_t\epsilon_t; \ \epsilon_t \sim \mathcal{N}(0, \mathbf{I}_n) \\ &= \frac{\sqrt{\boldsymbol{\nu}(t)}}{\sqrt{1 + \boldsymbol{\nu}(t)}}\mathbf{x}_0 + \frac{1}{\sqrt{1 + \boldsymbol{\nu}(t)}}\epsilon_t \qquad \text{Using Eq. 25} \end{aligned} \tag{26}$$

## B.2 REVERSE PROCESS

The distribution of $\mathbf{x}_t$ given $\mathbf{x}_s$ is given by:

$$q(\mathbf{x}_s|\mathbf{x}_t, \mathbf{x}_0) = \mathcal{N}\left(\boldsymbol{\mu}_q = \frac{\boldsymbol{\alpha}_{t|s}\boldsymbol{\sigma}_s^2}{\boldsymbol{\sigma}_t^2}\mathbf{x}_t + \frac{\boldsymbol{\sigma}_{t|s}^2\boldsymbol{\alpha}_s}{\boldsymbol{\sigma}_t^2}\mathbf{x}_0, \ \boldsymbol{\Sigma}_q = \text{diag}\left(\frac{\boldsymbol{\sigma}_s^2\boldsymbol{\sigma}_{t|s}^2}{\boldsymbol{\sigma}_t^2}\right)\right) \tag{27}$$

Let $\mathbf{x}_\theta(\mathbf{x}_t, t)$ be the neural network approximation for $\mathbf{x}_0$. Then we get the following reverse process:

$$p_\theta(\mathbf{x}_s|\mathbf{x}_t) = \mathcal{N}\left(\boldsymbol{\mu}_p = \frac{\boldsymbol{\alpha}_{t|s}\boldsymbol{\sigma}_s^2}{\boldsymbol{\sigma}_t^2}\mathbf{x}_t + \frac{\boldsymbol{\sigma}_{t|s}^2\boldsymbol{\alpha}_s}{\boldsymbol{\sigma}_t^2}\mathbf{x}_\theta(\mathbf{x}_t, t), \ \boldsymbol{\Sigma}_p = \text{diag}\left(\frac{\boldsymbol{\sigma}_s^2\boldsymbol{\sigma}_{t|s}^2}{\boldsymbol{\sigma}_t^2}\right)\right) \tag{28}$$

### B.3 DIFFUSION LOSS

For brevity we use the notation $s$ for $s(i)$ and $t$ for $t(i)$. From Eq. 27 and Eq. 28 we get the following expression for $q(\mathbf{x}_s|\mathbf{x}_t, \mathbf{x}_0)$:

$$D_{\mathrm{KL}}(q(\mathbf{x}_s|\mathbf{x}_t, \mathbf{x}_0)\|p_\theta(\mathbf{x}_s|\mathbf{x}_t))$$

$$= \frac{1}{2}\left((\boldsymbol{\mu}_q - \boldsymbol{\mu}_p)^\top \boldsymbol{\Sigma}_\theta^{-1}(\boldsymbol{\mu}_q - \boldsymbol{\mu}_p) + \mathrm{tr}\left(\boldsymbol{\Sigma}_q\boldsymbol{\Sigma}_p^{-1} - \mathbf{I}_n\right) - \log\frac{|\boldsymbol{\Sigma}_q|}{|\boldsymbol{\Sigma}_p|}\right)$$

$$= \frac{1}{2}(\boldsymbol{\mu}_q - \boldsymbol{\mu}_p)^\top \Sigma_\theta^{-1}(\boldsymbol{\mu}_q - \boldsymbol{\mu}_p)$$

Substituting $\boldsymbol{\mu}_q, \boldsymbol{\mu}_p, \boldsymbol{\Sigma}_p$ from equation 28 and equation 27.

$$= \frac{1}{2}\left(\frac{\boldsymbol{\sigma}_{t|s}^2\boldsymbol{\alpha}_s}{\boldsymbol{\sigma}_t^2}\mathbf{x}_0 - \frac{\boldsymbol{\sigma}_{t|s}^2\boldsymbol{\alpha}_s}{\boldsymbol{\sigma}_t^2}\mathbf{x}_\theta(\mathbf{x}_t,t)\right)^\top \mathrm{diag}\left(\frac{\boldsymbol{\sigma}_s^2\boldsymbol{\sigma}_{t|s}^2}{\boldsymbol{\sigma}_t^2}\right)^{-1}\left(\frac{\boldsymbol{\sigma}_{t|s}^2\boldsymbol{\alpha}_s}{\boldsymbol{\sigma}_t^2}\mathbf{x}_0 - \frac{\boldsymbol{\sigma}_{t|s}^2\boldsymbol{\alpha}_s}{\boldsymbol{\sigma}_t^2}\mathbf{x}_\theta(\mathbf{x}_t,t)\right)$$

$$= \frac{1}{2}(\mathbf{x}_0 - \mathbf{x}_\theta(\mathbf{x}_t,t))^\top \mathrm{diag}\left(\frac{\boldsymbol{\sigma}_{t|s}^2\boldsymbol{\alpha}_s}{\boldsymbol{\sigma}_t^2}\right)^\top \mathrm{diag}\left(\frac{\boldsymbol{\sigma}_s^2\boldsymbol{\sigma}_{t|s}^2}{\boldsymbol{\sigma}_t^2}\right)^{-1}\mathrm{diag}\left(\frac{\boldsymbol{\sigma}_{t|s}^2\boldsymbol{\alpha}_s}{\boldsymbol{\sigma}_t^2}\right)(\mathbf{x}_0 - \mathbf{x}_\theta(\mathbf{x}_t,t))$$

$$= \frac{1}{2}(\mathbf{x}_0 - \mathbf{x}_\theta(\mathbf{x}_t,t))^\top \mathrm{diag}\left(\frac{\boldsymbol{\sigma}_{t|s}^2\boldsymbol{\alpha}_s}{\boldsymbol{\sigma}_t^2}\odot\frac{\boldsymbol{\sigma}_t^2}{\boldsymbol{\sigma}_s^2\boldsymbol{\sigma}_{t|s}^2}\odot\frac{\boldsymbol{\sigma}_{t|s}^2\boldsymbol{\alpha}_s}{\boldsymbol{\sigma}_t^2}\right)(\mathbf{x}_0 - \mathbf{x}_\theta(\mathbf{x}_t,t))$$

$$= \frac{1}{2}(\mathbf{x}_0 - \mathbf{x}_\theta(\mathbf{x}_t,t))^\top \mathrm{diag}\left(\frac{\boldsymbol{\sigma}_{t|s}^2\boldsymbol{\alpha}_s^2}{\boldsymbol{\sigma}_t^2\boldsymbol{\sigma}_s^2}\right)(\mathbf{x}_0 - \mathbf{x}_\theta(\mathbf{x}_t,t))$$

Simplifying the expression using eq. 22 and eq. 23 we get,

$$= \frac{1}{2}(\mathbf{x}_0 - \mathbf{x}_\theta(\mathbf{x}_t,t))^\top \mathrm{diag}\left(\frac{\boldsymbol{\alpha}_s^2}{\boldsymbol{\sigma}_s^2} - \frac{\boldsymbol{\alpha}_t^2}{\boldsymbol{\sigma}_t^2}\right)(\mathbf{x}_0 - \mathbf{x}_\theta(\mathbf{x}_t,t))$$

Using the relation $\boldsymbol{\nu}(t) = \boldsymbol{\alpha}_t^2/\boldsymbol{\sigma}_t^2$ we get,

$$= \frac{1}{2}(\mathbf{x}_0 - \mathbf{x}_\theta(\mathbf{x}_t,t))^\top \mathrm{diag}\left(\boldsymbol{\nu}(s) - \boldsymbol{\nu}(t)\right)(\mathbf{x}_0 - \mathbf{x}_\theta(\mathbf{x}_t,t)) \tag{29}$$

Like Kingma et al. (2021) we train the model in the continuous domain with $T \to \infty$.

$$\mathcal{L}_{\mathrm{diffusion}}$$

$$= \lim_{T\to\infty}\frac{1}{2}\sum_{i=2}^{T}\mathbb{E}_{\epsilon\sim\mathcal{N}(0,\mathbf{I}_n)}\mathrm{D}_{\mathrm{KL}}(q(\mathbf{x}_{s(i)}|\mathbf{x}_{t(i)},\mathbf{x}_0)\|p_\theta(\mathbf{x}_{s(i)}|\mathbf{x}_{t(i)}))$$

$$= \lim_{T\to\infty}\frac{1}{2}\sum_{i=2}^{T}\mathbb{E}_{\epsilon\sim\mathcal{N}(0,\mathbf{I}_n)}(\mathbf{x}_0 - \mathbf{x}_\theta(\mathbf{x}_{t(i)},t(i)))^\top \mathrm{diag}\left(\boldsymbol{\nu}_{s(i)} - \boldsymbol{\nu}_{t(i)}\right)(\mathbf{x}_0 - \mathbf{x}_\theta(\mathbf{x}_{t(i)},t))$$

$$= \frac{1}{2}\mathbb{E}_{\epsilon\sim\mathcal{N}(0,\mathbf{I}_n)}\left[\lim_{T\to\infty}\sum_{i=2}^{T}(\mathbf{x}_0 - \mathbf{x}_\theta(\mathbf{x}_{t(i)},t(i)))^\top \mathrm{diag}\left(\boldsymbol{\nu}_{s(i)} - \boldsymbol{\nu}_{t(i)}\right)(\mathbf{x}_0 - \mathbf{x}_\theta(\mathbf{x}_{t(i)},t))\right]$$

$$= \frac{1}{2}\mathbb{E}_{\epsilon\sim\mathcal{N}(0,\mathbf{I}_n)}\left[\lim_{T\to\infty}\sum_{i=2}^{T}T(\mathbf{x}_0 - \mathbf{x}_\theta(\mathbf{x}_{t(i)},t(i)))^\top \mathrm{diag}\left(\boldsymbol{\nu}_{s(i)} - \boldsymbol{\nu}_{t(i)}\right)(\mathbf{x}_0 - \mathbf{x}_\theta(\mathbf{x}_{t(i)},t))\frac{1}{T}\right]$$

Let $\lim_{T\to\infty}T(\boldsymbol{\nu}_{s(i)} - \boldsymbol{\nu}_{t(i)}) = \frac{d}{dt}\boldsymbol{\nu}(t)$ denote the scalar derivative of the vector $\boldsymbol{\nu}(t)$ w.r.t $t$

$$= \frac{1}{2}\mathbb{E}_{\epsilon\sim\mathcal{N}(0,\mathbf{I}_n)}\left[\int_0^1 (\mathbf{x}_0 - \mathbf{x}_\theta(\mathbf{x}_t,t))^\top \mathrm{diag}\left(\frac{d}{dt}\boldsymbol{\nu}(t)\right)(\mathbf{x}_0 - \mathbf{x}_\theta(\mathbf{x}_t,t))dt\right] \tag{30}$$

In practice instead of computing the integral is computed by MC sampling.

$$= -\frac{1}{2}\mathbb{E}_{\epsilon\sim\mathcal{N}(0,\mathbf{I}_n),t\sim U[0,1]}\left[(\mathbf{x}_0 - \mathbf{x}_\theta(\mathbf{x}_t,t))^\top \mathrm{diag}\left(\frac{d}{dt}\boldsymbol{\nu}(t)\right)(\mathbf{x}_0 - \mathbf{x}_\theta(\mathbf{x}_t,t))\right] \tag{31}$$

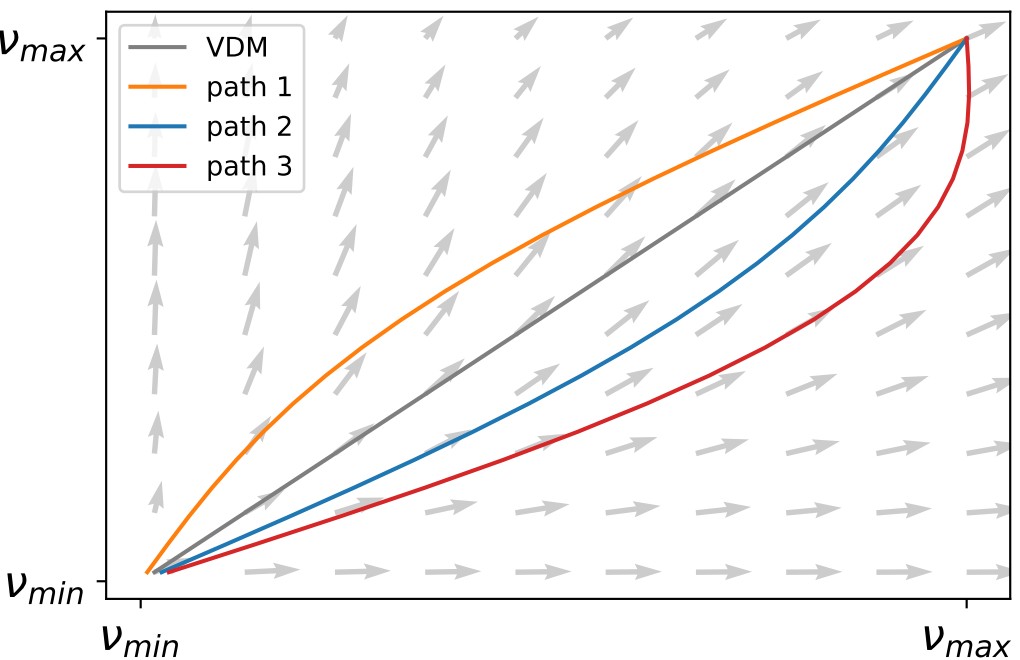

Figure 3: Caption

## B.4 Vectorized Representation of the diffusion loss

Let $\boldsymbol{\nu}(t)$ be the vectorized representation of the diagonal entries of the matrix $\boldsymbol{\nu}(t)$. We can rewrite the integral in eq. 30 in the following vectorized form where $\odot$ denotes element wise multiplication and $\langle , \rangle$ denotes dot product between 2 vectors.

$$\int_0^1 (\mathbf{x}_0 - \mathbf{x}_\theta(\mathbf{x}_t, t))^\top \mathrm{diag}\left(\frac{d}{dt}\boldsymbol{\nu}(t)\right)(\mathbf{x}_0 - \mathbf{x}_\theta(\mathbf{x}_t, t))dt$$

$$= -\frac{1}{2}\int_0^1 \langle (\mathbf{x}_0 - \mathbf{x}_\theta(\mathbf{x}_t, t)) \odot (\mathbf{x}_0 - \mathbf{x}_\theta(\mathbf{x}_t, t)), \frac{d}{dt}\boldsymbol{\nu}(t)\rangle dt$$

Using change of variables as mentioned in Sec. 3.2 we have

$$= -\frac{1}{2}\int_0^1 \langle (\mathbf{x}_0 - \tilde{\mathbf{x}}_\theta(\mathbf{x}_{\boldsymbol{\nu}(t)}, \boldsymbol{\nu}(t))) \odot (\mathbf{x}_0 - \tilde{\mathbf{x}}_\theta(\mathbf{x}_{\boldsymbol{\nu}(t)}, \boldsymbol{\nu}(t))), \frac{d}{dt}\boldsymbol{\nu}(t)\rangle dt$$

Let $\mathbf{f}_\theta(\mathbf{x}_0, \boldsymbol{\nu}(t)) = (\mathbf{x}_0 - \tilde{\mathbf{x}}_\theta(\mathbf{x}_{\boldsymbol{\nu}(t)}, \boldsymbol{\nu}(t))) \odot (\mathbf{x}_0 - \tilde{\mathbf{x}}_\theta(\mathbf{x}_{\boldsymbol{\nu}(t)}, \boldsymbol{\nu}(t)))$

$$= \int_0^1 \langle \mathbf{f}_\theta(\mathbf{x}_0, \boldsymbol{\nu}(t)), \frac{d}{dt}\boldsymbol{\nu}(t)\rangle dt \tag{32}$$

Thus $\mathcal{L}_{\text{diffusion}}$ can be interpreted as the amount of work done along the trajectory $\boldsymbol{\nu}(0) \to \boldsymbol{\nu}(1)$ in the presence of a vector field $\mathbf{f}_\theta(\mathbf{x}_0, \boldsymbol{\nu}(\mathbf{z}, t))$. From the perspective of thermodynamics, this is precisely equal to the amount of heat lost into the environment during the process of transition between 2 equilibria via the noise schedule specified by $\boldsymbol{\nu}(t)$.

TODO

## B.5 Log likelihood and Noise Schedules: A Thermodynamics perspective

A diffusion model characterizes a quasi-static process that occurs between two equilibrium distributions: $q(\mathbf{x}_0) \to q(x_1)$, via a stochastic trajectory (Sohl-Dickstein et al., 2015). According to

Spinney & Ford (2012), it is demonstrated that the diffusion schedule or the noising process plays a pivotal role in determining the "measure of irreversibility" for this stochastic trajectory which is expressed as $\log \frac{P_F(\mathbf{x}_{0:1})}{P_B(\mathbf{x}_{1:0})}$. $P_F(\mathbf{x}_{0:1})$ represents the probability of observing the forward path $\mathbf{x}_{0:1}$ and $P_B(\mathbf{x}_{1:0})$ represents the probability of observing the reverse path $\mathbf{x}_{1:0}$. It's worth noting that $\log \frac{P_F(\mathbf{x}_{0:1})}{P_B(\mathbf{x}_{1:0})}$ corresponds precisely to the ELBO Eq. 1 that we optimize when training a diffusion model. Consequently, thermodynamics asserts that the noise schedule indeed has an impact on the log-likelihood of the diffusion model which contradicts Kingma et al. (2021).

## C  MULTIVARIATE NOISE SCHEDULE CONDITIONED ON CONTEXT

Let's say we have a context variable $\mathbf{c} \in \mathbb{R}^m$ that captures high level information about $\mathbf{x}_0$. $\boldsymbol{\alpha}_t(\mathbf{c}), \boldsymbol{\sigma}_t(\mathbf{c}) \in \mathbb{R}^{n \times n}$ are diagonal matrices. The timesteps $s, t$ satisfy $0 \leq s < t \leq 1$. Furthermore, we use the following notations:

$$\boldsymbol{\alpha}_{t|s}(\mathbf{c}) = \boldsymbol{\alpha}_t(\mathbf{c})\boldsymbol{\alpha}_s^{-1}(\mathbf{c}) \tag{33}$$

$$\boldsymbol{\sigma}_{t|s}^2(\mathbf{c}) = \boldsymbol{\sigma}_t^2(\mathbf{c}) - \boldsymbol{\alpha}_{t|s}^2(\mathbf{c})\boldsymbol{\sigma}_s^{-2}(\mathbf{c}) \tag{34}$$

The forward process for such a method is given as:

$$q_\phi(\mathbf{x}_t|\mathbf{x}_s, \mathbf{c}) = \mathcal{N}\left(\boldsymbol{\alpha}_{t|s}(\mathbf{c})\mathbf{x}_s, \boldsymbol{\sigma}_{t|s}^2(\mathbf{c})\right) \tag{35}$$

The distribution of $\mathbf{x}_t$ given $\mathbf{x}_s$ is given by (the derivation is similar to Hoogeboom & Salimans (2022)):

$$q_\phi(\mathbf{x}_s|\mathbf{x}_t, \mathbf{x}_0, \mathbf{c})$$
$$= \mathcal{N}\left(\boldsymbol{\mu}_q = \frac{\boldsymbol{\alpha}_{t|s}(\mathbf{c})\boldsymbol{\sigma}_s^2(\mathbf{c})}{\boldsymbol{\sigma}_t^2(\mathbf{c})}\mathbf{x}_t + \frac{\boldsymbol{\sigma}_{t|s}^2(\mathbf{c})\boldsymbol{\alpha}_s(\mathbf{c})}{\boldsymbol{\sigma}_t^2(\mathbf{c})}\mathbf{x}_0, \ \boldsymbol{\Sigma}_q = \operatorname{diag}\left(\frac{\boldsymbol{\sigma}_s^2(\mathbf{c})\boldsymbol{\sigma}_{t|s}^2(\mathbf{c})}{\boldsymbol{\sigma}_t^2(\mathbf{c})}\right)\right) \tag{36}$$

### C.1  CONTEXT IS AVAILABLE DURING THE INFERENCE TIME.

Even though $\mathbf{c}$ represents the input $\mathbf{x}_0$, it could be available during during inference. For example $\mathbf{c}$ could be class labels (Dhariwal & Nichol, 2021) or prexisting embeddings from an auto-encoder (Preechakul et al., 2022).

#### C.1.1  REVERSE PROCESS: APPROXIMATE

Let $\mathbf{x}_\theta(\mathbf{x}_t, \mathbf{c}; t)$ be an approximation for $\mathbf{x}_0$. Then we get the following reverse process (for brevity we write $\mathbf{x}_\theta(\mathbf{x}_t, \mathbf{c}; t)$ as $\mathbf{x}_\theta$):

$$p_\theta(\mathbf{x}_s|\mathbf{x}_t, \mathbf{c}) = \mathcal{N}\left(\boldsymbol{\mu}_p = \frac{\boldsymbol{\alpha}_{t|s}(\mathbf{c})\boldsymbol{\sigma}_s^2(\mathbf{c})}{\boldsymbol{\sigma}_t^2(\mathbf{c})}\mathbf{x}_t + \frac{\boldsymbol{\sigma}_{t|s}^2(\mathbf{c})\boldsymbol{\alpha}_s(\mathbf{c})}{\boldsymbol{\sigma}_t^2(\mathbf{c})}\mathbf{x}_\theta, \ \boldsymbol{\Sigma}_p = \operatorname{diag}\left(\frac{\boldsymbol{\sigma}_s^2(\mathbf{c})\boldsymbol{\sigma}_{t|s}^2(\mathbf{c})}{\boldsymbol{\sigma}_t^2(\mathbf{c})}\right)\right) \tag{37}$$

#### C.1.2  DIFFUSION LOSS

Similar to the derivation of multi-variate $\mathcal{L}_{\text{diffusion}}$ in Eq. 29 we can derive $\mathcal{L}_{\text{diffusion}}$ for this case too:

$$\mathcal{L}_{\text{diffusion}} = -\frac{1}{2}\mathbb{E}_{\epsilon \sim \mathcal{N}(0, \mathbf{I}_n), t \sim U[0,1]}\left[(\mathbf{x}_0 - \mathbf{x}_\theta(\mathbf{x}_t, \mathbf{c}; t))^\top \operatorname{diag}\left(\frac{d}{dt}\boldsymbol{\nu}(t)\right)(\mathbf{x}_0 - \mathbf{x}_\theta(\mathbf{x}_t, \mathbf{c}; t))\right] \tag{38}$$

#### C.1.3  LIMITATIONS OF THIS METHOD

This approach is very limited where the diffusion process is only conditioned on class labels. Using pre-existing embeddings like Diff-AE (Preechakul et al., 2022) is also not possible in general and is only limited to tasks such as attribute manipulation in datasets.

### C.2 CONTEXT **ISN'T** AVAILABLE DURING THE INFERENCE TIME.

If the context, $\mathbf{c}$ is an explicit function of the input $\mathbf{x}_0$ things become challenging because $\mathbf{x}_0$ isn't available during the inference stage. For this reason, Eq. 36 can't be used to parameterize $\boldsymbol{\mu}_p, \boldsymbol{\Sigma}_p$ in $p_\theta(\mathbf{x}_s|\mathbf{x}_t)$. Let $p_\theta(\mathbf{x}_s|\mathbf{x}_t) = \mathcal{N}(\boldsymbol{\mu}_p(\mathbf{x}_t, t), \boldsymbol{\Sigma}_p(\mathbf{x}_t, t))$ where $\boldsymbol{\mu}_p, \boldsymbol{\Sigma}_p$ are parameterized directly by a neural network. Using Eq. 4 we get the following diffusion loss:

$$\mathcal{L}_{\text{diffusion}} = T\,\mathbb{E}_{i \sim U[0,T]} D_{\text{KL}}\left(q(\mathbf{x}_{s(i)}|\mathbf{x}_{t(i)}, \mathbf{x}_0) \| p_\theta(\mathbf{x}_{s(i)}|\mathbf{x}_{t(i)})\right)$$

$$= \mathbb{E}_{q_\phi}\left(\underbrace{\frac{T}{2}(\boldsymbol{\mu}_q - \boldsymbol{\mu}_p)^\top \boldsymbol{\Sigma}_\theta^{-1}(\boldsymbol{\mu}_q - \boldsymbol{\mu}_p)}_{\text{term 1}} + \underbrace{\frac{T}{2}\left(\text{tr}\left(\boldsymbol{\Sigma}_q \boldsymbol{\Sigma}_p^{-1} - \mathbf{I}_n\right) - \log \frac{|\boldsymbol{\Sigma}_q|}{|\boldsymbol{\Sigma}_p|}\right)}_{\text{term 2}}\right) \quad (39)$$

#### C.2.1 REVERSE PROCESS: APPROXIMATE

Due to the challenges associated with parameterizing $\boldsymbol{\mu}_p, \boldsymbol{\Sigma}_p$ directly using a neural network we parameterize $\mathbf{c}$ using a neural network that approximates $\mathbf{c}$ in the reverse process. Let $\mathbf{x}_\theta(\mathbf{x}_t, t)$ be an approximation for $\mathbf{x}_0$. Then we get the following reverse Rrocess (for brevity we write $\mathbf{x}_\theta(\mathbf{x}_t, t)$ as $\mathbf{x}_\theta$, and $\mathbf{c}_\theta$ denotes an approximation to $\mathbf{c}$ in the reverse process.):

$$p_\theta(\mathbf{x}_s|\mathbf{x}_t)$$
$$= \mathcal{N}\left(\boldsymbol{\mu}_p = \frac{\boldsymbol{\alpha}_{t|s}(\mathbf{c}_\theta)\boldsymbol{\sigma}_s^2(\mathbf{c}_\theta)}{\boldsymbol{\sigma}_t^2(\mathbf{c}_\theta)}\mathbf{x}_t + \frac{\boldsymbol{\sigma}_{t|s}^2(\mathbf{c}_\theta)\boldsymbol{\alpha}_s(\mathbf{c}_\theta)}{\boldsymbol{\sigma}_t^2(\mathbf{c}_\theta)}\mathbf{x}_\theta,\ \boldsymbol{\Sigma}_p = \text{diag}\left(\frac{\boldsymbol{\sigma}_s^2(\mathbf{c}_\theta)\boldsymbol{\sigma}_{t|s}^2(\mathbf{c}_\theta)}{\boldsymbol{\sigma}_t^2(\mathbf{c}_\theta)}\right)\right) \quad (40)$$

Consider the limiting case where $T \to \infty$. Let's analyze the 2 terms in Eq. 39 separately.

Using Eq. 4 and Eq. 6, **term 1** in Eq. 39 simplifies in the following manner:

$$\lim_{T \to \infty} \frac{T}{2}(\boldsymbol{\mu}_q - \boldsymbol{\mu}_p)^\top \boldsymbol{\Sigma}_\theta^{-1}(\boldsymbol{\mu}_q - \boldsymbol{\mu}_p)$$

$$\lim_{T \to \infty} \frac{T}{2} \sum_{i=1}^d \frac{((\boldsymbol{\mu}_q)_i - (\boldsymbol{\mu}_p)_i)^2}{(\boldsymbol{\Sigma}_\theta)_i} \quad (41)$$

Substituting 1 / T as $\delta$

$$\lim_{\delta \to 0^+} \sum_{i=1}^d \frac{1}{\delta \boldsymbol{\sigma}_i^2(\mathbf{x}_\theta, t - \delta)\left(1 - \frac{\boldsymbol{\nu}_i(\mathbf{x}_\theta, t)}{\boldsymbol{\nu}_i(\mathbf{x}_\theta, t - \delta)}\right)} \times$$

$$\left[\frac{\boldsymbol{\alpha}_i(\mathbf{x}, t - \delta)}{\boldsymbol{\alpha}_i(\mathbf{x}, t)} \frac{\boldsymbol{\nu}_i(\mathbf{x}, t)}{\boldsymbol{\nu}_i(\mathbf{x}, t - \delta)}\mathbf{z}_t + \boldsymbol{\alpha}_i(\mathbf{x}, t - \delta)\left(1 - \frac{\boldsymbol{\nu}_i(\mathbf{x}, t)}{\boldsymbol{\nu}_i(\mathbf{x}, t - \delta)}\right)x_i\right.$$

$$\left. - \frac{\boldsymbol{\alpha}_i(\mathbf{x}_\theta, t - \delta)}{\boldsymbol{\alpha}_i(\mathbf{x}_\theta, t)} \frac{\boldsymbol{\nu}_i(\mathbf{x}_\theta, t)}{\boldsymbol{\nu}_i(\mathbf{x}_\theta, t - \delta)}\mathbf{z}_t + \boldsymbol{\alpha}_i(\mathbf{x}_\theta, t - \delta)\left(1 - \frac{\boldsymbol{\nu}_i(\mathbf{x}_\theta, t)}{\boldsymbol{\nu}_i(\mathbf{x}_\theta, t - \delta)}\right)(x_\theta)_i\right]^2 \quad (42)$$

Consider the scalar case: substituting $\delta = 1/T$,

$$\lim_{\delta \to 0} \frac{1}{\delta \sigma^2(\mathbf{x}_\theta, t - \delta)\left(1 - \frac{\nu(\mathbf{x}_\theta, t)}{\nu(\mathbf{x}_\theta, t - \delta)}\right)} \times$$

$$\left[\frac{\alpha(\mathbf{x}, t - \delta)}{\alpha(\mathbf{x}, t)} \frac{\nu(\mathbf{x}, t)}{\nu(\mathbf{x}, t - \delta)}\mathbf{z}_t + \alpha(\mathbf{x}, t - \delta)\left(1 - \frac{\nu(\mathbf{x}, t)}{\nu(\mathbf{x}, t - \delta)}\right)\mathbf{x}\right.$$

$$\left. - \frac{\alpha(\mathbf{x}_\theta, t - \delta)}{\alpha(\mathbf{x}_\theta, t)} \frac{\nu(\mathbf{x}_\theta, t)}{\nu(\mathbf{x}_\theta, t - \delta)}\mathbf{z}_t + \alpha(\mathbf{x}_\theta, t - \delta)\left(1 - \frac{\nu(\mathbf{x}_\theta, t)}{\nu(\mathbf{x}_\theta, t - \delta)}\right)\mathbf{x}_\theta\right]^2 \quad (43)$$

Notice that this equation is in indeterminate for when we substitute $\delta = 0$. One can apply L'Hospital rule twice or break it down into 3 terms below. For this reason let's write it as

$$\text{expression 1: } \lim_{\delta \to 0} \frac{1}{\delta} \times \left[ \frac{\alpha(\mathbf{x}, t - \delta)}{\alpha(\mathbf{x}, t)} \frac{\nu(\mathbf{x}, t)}{\nu(\mathbf{x}, t - \delta)} \mathbf{z}_t + \alpha(\mathbf{x}, t - \delta) \left( 1 - \frac{\nu(\mathbf{x}, t)}{\nu(\mathbf{x}, t - \delta)} \right) \mathbf{x} \right.$$

$$\left. - \frac{\alpha(\mathbf{x}_\theta, t - \delta)}{\alpha(\mathbf{x}_\theta, t)} \frac{\nu(\mathbf{x}_\theta, t)}{\nu(\mathbf{x}_\theta, t - \delta)} \mathbf{z}_t + \alpha(\mathbf{x}_\theta, t - \delta) \left( 1 - \frac{\nu(\mathbf{x}_\theta, t)}{\nu(\mathbf{x}_\theta, t - \delta)} \right) \mathbf{x}_\theta \right] \tag{44}$$

$$\text{expression 2: } \lim_{\delta \to 0} \frac{1}{\left( 1 - \frac{\nu(\mathbf{x}_\theta, t)}{\nu(\mathbf{x}_\theta, t - \delta)} \right)} \times \left[ \frac{\alpha(\mathbf{x}, t - \delta)}{\alpha(\mathbf{x}, t)} \frac{\nu(\mathbf{x}, t)}{\nu(\mathbf{x}, t - \delta)} \mathbf{z}_t + \alpha(\mathbf{x}, t - \delta) \left( 1 - \frac{\nu(\mathbf{x}, t)}{\nu(\mathbf{x}, t - \delta)} \right) \mathbf{x} \right.$$

$$\left. - \frac{\alpha(\mathbf{x}_\theta, t - \delta)}{\alpha(\mathbf{x}_\theta, t)} \frac{\nu(\mathbf{x}_\theta, t)}{\nu(\mathbf{x}_\theta, t - \delta)} \mathbf{z}_t + \alpha(\mathbf{x}_\theta, t - \delta) \left( 1 - \frac{\nu(\mathbf{x}_\theta, t)}{\nu(\mathbf{x}_\theta, t - \delta)} \right) \mathbf{x}_\theta \right]^2 \tag{45}$$

Applying L'Hospital rule in expression 1 we get,

$$\frac{d}{d\delta} \left( \frac{\alpha(\mathbf{x}, t - \delta)}{\alpha(\mathbf{x}, t)} \frac{\nu(\mathbf{x}, t)}{\nu(\mathbf{x}, t - \delta)} \right) \Bigg|_{\delta = 0} = \frac{\nu(\mathbf{x}, t)}{\alpha(\mathbf{x}, t)} \frac{-\nu(\mathbf{x}, t)\alpha'(\mathbf{x}, t) + \alpha(\mathbf{x}, t)\nu'(\mathbf{x}, t)}{\nu^2(\mathbf{x}, t)}$$

$$= \frac{-\alpha'(\mathbf{x}, t)}{\alpha(\mathbf{x}, t)} + \frac{\nu'(\mathbf{x}, t)}{\nu(\mathbf{x}, t)} \tag{46}$$

$$\frac{d}{d\delta} \alpha(\mathbf{x}, t - \delta) \left( 1 - \frac{\nu(\mathbf{x}, t)}{\nu(\mathbf{x}, t - \delta)} \right) \Bigg|_{\delta = 0} = -\alpha(\mathbf{x}, t) \frac{\nu'(\mathbf{x}, t)}{\nu(\mathbf{x}, t)} \tag{47}$$

$$\left[ \left( \frac{-\alpha'(\mathbf{x}, t)}{\alpha(\mathbf{x}, t)} + \frac{\nu'(\mathbf{x}, t)}{\nu(\mathbf{x}, t)} + \frac{\alpha'(\mathbf{x}_\theta, t)}{\alpha(\mathbf{x}_\theta, t)} - \frac{\nu'(\mathbf{x}_\theta, t)}{\nu(\mathbf{x}_\theta, t)} \right) \mathbf{z}_t \tag{48} \right.$$

$$\left. -\alpha(\mathbf{x}, t) \frac{\nu'(\mathbf{x}, t)}{\nu(\mathbf{x}, t)} \mathbf{x} + \alpha(\mathbf{x}_\theta, t) \frac{\nu'(\mathbf{x}_\theta, t)}{\nu(\mathbf{x}_\theta, t)} \mathbf{x}_\theta \right]^2 \times \frac{\nu(\mathbf{x}, t)}{\nu'(\mathbf{x}, t)} \tag{49}$$

Thus the final result:

$$\sum_{i=1}^{d} \left[ \left( \frac{-\boldsymbol{\alpha}_i'(\mathbf{x}, t)}{\boldsymbol{\alpha}_i(\mathbf{x}, t)} + \frac{\boldsymbol{\nu}_i'(\mathbf{x}, t)}{\boldsymbol{\nu}_i(\mathbf{x}, t)} + \frac{\boldsymbol{\alpha}_i'(\mathbf{x}_\theta, t)}{\boldsymbol{\alpha}_i(\mathbf{x}_\theta, t)} - \frac{\boldsymbol{\nu}_i'(\mathbf{x}_\theta, t)}{\boldsymbol{\nu}_i(\mathbf{x}_\theta, t)} \right) \mathbf{z}_t \right.$$

$$\left. - \boldsymbol{\alpha}_i(\mathbf{x}, t) \frac{\boldsymbol{\nu}_i'(\mathbf{x}, t)}{\boldsymbol{\nu}_i(\mathbf{x}, t)} \mathbf{x} + \boldsymbol{\alpha}_i(\mathbf{x}_\theta, t) \frac{\boldsymbol{\nu}_i'(\mathbf{x}_\theta, t)}{\boldsymbol{\nu}_i(\mathbf{x}_\theta, t)} \mathbf{x}_\theta \right]^2 \times \frac{\boldsymbol{\nu}_i(\mathbf{x}, t)}{\boldsymbol{\nu}_i'(\mathbf{x}, t)}$$

$$= \Lambda^\top \text{diag} \left( \frac{\boldsymbol{\nu}(\mathbf{x}, t)}{\boldsymbol{\nu}'(\mathbf{x}, t)} \right) \Lambda$$

$$\text{where } \Lambda = \left[ \left( \frac{-\boldsymbol{\alpha}'(\mathbf{x}, t)}{\boldsymbol{\alpha}(\mathbf{x}, t)} + \frac{\boldsymbol{\nu}'(\mathbf{x}, t)}{\boldsymbol{\nu}(\mathbf{x}, t)} + \frac{\boldsymbol{\alpha}'(\mathbf{x}_\theta, t)}{\boldsymbol{\alpha}(\mathbf{x}_\theta, t)} - \frac{\boldsymbol{\nu}'(\mathbf{x}_\theta, t)}{\boldsymbol{\nu}(\mathbf{x}_\theta, t)} \right) \mathbf{z}_t - \boldsymbol{\alpha}(\mathbf{x}, t) \frac{\boldsymbol{\nu}'(\mathbf{x}, t)}{\boldsymbol{\nu}(\mathbf{x}, t)} \mathbf{x} + \boldsymbol{\alpha}(\mathbf{x}_\theta, t) \frac{\boldsymbol{\nu}'(\mathbf{x}_\theta, t)}{\boldsymbol{\nu}(\mathbf{x}_\theta, t)} \mathbf{x}_\theta \right] \tag{50}$$

For the second term we have the following:

$$\lim_{T \to \infty} \frac{T}{2} \left( \text{tr} \left( \boldsymbol{\Sigma}_q \boldsymbol{\Sigma}_p^{-1} - \mathbf{I}_n \right) - \log \frac{|\boldsymbol{\Sigma}_q|}{|\boldsymbol{\Sigma}_p|} \right)$$

steps todo

$$\lim_{T \to \infty} \frac{T}{2} \left[ \text{tr} \left( \text{diag} \left( \boldsymbol{\sigma}^2(\mathbf{c}, s) \left( 1 - \frac{\boldsymbol{\nu}(\mathbf{c}, t)}{\boldsymbol{\nu}(\mathbf{c}, s)} \right) \right) \Big/ \text{diag} \left( \boldsymbol{\sigma}^2(\mathbf{c}_\theta, s) \left( 1 - \frac{\boldsymbol{\nu}(\mathbf{c}_\theta, t)}{\boldsymbol{\nu}(\mathbf{c}_\theta, s)} \right) \right) - \mathbf{I}_n \right) \right.$$
$$\left. - \log \frac{\left| \text{diag} \left( \boldsymbol{\sigma}^2(\mathbf{c}, s)(1 - \frac{\boldsymbol{\nu}(\mathbf{c}, t)}{\boldsymbol{\nu}(\mathbf{c}, s)}) \right) \right|}{\left| \text{diag} \left( \boldsymbol{\sigma}^2(\mathbf{c}_\theta, s)(1 - \frac{\boldsymbol{\nu}(\mathbf{c}_\theta, t)}{\boldsymbol{\nu}(\mathbf{c}_\theta, s)}) \right) \right|} \right]$$

$$\lim_{T \to \infty} \frac{T}{2} \sum_{i=1}^{d} \left( \frac{\boldsymbol{\sigma}_i^2(\mathbf{c}, s) \left( 1 - \frac{\boldsymbol{\nu}_i(\mathbf{c}, t)}{\boldsymbol{\nu}_i(\mathbf{c}, s)} \right)}{\boldsymbol{\sigma}_i^2(\mathbf{c}_\theta, s) \left( 1 - \frac{\boldsymbol{\nu}_i(\mathbf{c}_\theta, t)}{\boldsymbol{\nu}_i(\mathbf{c}_\theta, s)} \right)} - 1 - \log \frac{\boldsymbol{\sigma}_i^2(\mathbf{c}, s) \left( 1 - \frac{\boldsymbol{\nu}_i(\mathbf{c}, t)}{\boldsymbol{\nu}_i(\mathbf{c}, s)} \right)}{\boldsymbol{\sigma}_i^2(\mathbf{c}_\theta, s) \left( 1 - \frac{\boldsymbol{\nu}_i(\mathbf{c}_\theta, t)}{\boldsymbol{\nu}_i(\mathbf{c}_\theta, s)} \right)} \right) \tag{51}$$

$$\tag{52}$$

Let $p_i = \frac{\boldsymbol{\sigma}_i^2(\mathbf{c}, s) \left( 1 - \frac{\boldsymbol{\nu}_i(\mathbf{c}, t)}{\boldsymbol{\nu}_i(\mathbf{c}, s)} \right)}{\boldsymbol{\sigma}_i^2(\mathbf{c}_\theta, s) \left( 1 - \frac{\boldsymbol{\nu}_i(\mathbf{c}_\theta, t)}{\boldsymbol{\nu}_i(\mathbf{c}_\theta, s)} \right)}$

The sequence $\lim_{T \to \infty} \frac{T}{2} \sum_{i=1}^{d} (p_i - 1 - \log p_i)$ converges iff $\lim_{T \to \infty} \sum_{i=1}^{d} (p_i - 1 - \log p_i) = 0$. Notice that the function $f(x) = x - 1 - \log x \geq 0 \ \forall x \in \mathbb{R}$ and the equality holds for $x = 1$. Thus, the condition $\lim_{T \to \infty} \frac{T}{2} \sum_{i=1}^{d} (p_i - 1 - \log p_i)$ holds iff $\lim_{T \to \infty} p_i = 0 \ \forall i \in \{1, \dots, d\}$. Thus,

$$\lim_{T \to \infty} p_i = 1$$

$$\implies \lim_{T \to \infty} \left( \frac{\boldsymbol{\sigma}_i^2(\mathbf{c}, s) \left( 1 - \frac{\boldsymbol{\nu}_i(\mathbf{c}, t)}{\boldsymbol{\nu}_i(\mathbf{c}, s)} \right)}{\boldsymbol{\sigma}_i^2(\mathbf{c}_\theta, s) \left( 1 - \frac{\boldsymbol{\nu}_i(\mathbf{c}_\theta, t)}{\boldsymbol{\nu}_i(\mathbf{c}_\theta, s)} \right)} \right) = 1$$

Substituting 1/T as $\delta$,

$$\implies \lim_{\delta \to 0^+} \left( \frac{\boldsymbol{\sigma}_i^2(\mathbf{c}, t - \delta) \left( 1 - \frac{\boldsymbol{\nu}_i(\mathbf{c}, t)}{\boldsymbol{\nu}_i(\mathbf{c}, t - \delta)} \right)}{\boldsymbol{\sigma}_i^2(\mathbf{c}_\theta, t - \delta) \left( 1 - \frac{\boldsymbol{\nu}_i(\mathbf{c}_\theta, t)}{\boldsymbol{\nu}_i(\mathbf{c}_\theta, t - \delta)} \right)} \right) = 1$$

$$\implies \frac{\boldsymbol{\sigma}_i^2(\mathbf{c}, t)}{\boldsymbol{\sigma}_i^2(\mathbf{c}_\theta, t)} \lim_{\delta \to 0^+} \left( \frac{1 - \frac{\boldsymbol{\nu}_i(\mathbf{c}, t)}{\boldsymbol{\nu}_i(\mathbf{c}, t - \delta)}}{1 - \frac{\boldsymbol{\nu}_i(\mathbf{c}_\theta, t)}{\boldsymbol{\nu}_i(\mathbf{c}_\theta, t - \delta)}} \right) = 1$$

Applying L'Hospital rule,

$$\implies \frac{\boldsymbol{\sigma}_i^2(\mathbf{c}, t)}{\boldsymbol{\sigma}_i^2(\mathbf{c}_\theta, t)} \left( \frac{\frac{-\boldsymbol{\nu}_i'(\mathbf{c}, t)}{\boldsymbol{\nu}_i(\mathbf{c}, t)}}{\frac{-\boldsymbol{\nu}_i'(\mathbf{c}_\theta, t)}{\boldsymbol{\nu}_i(\mathbf{c}_\theta, t)}} \right) = 1$$

$$\implies \frac{\boldsymbol{\sigma}_i^2(\mathbf{c}, t)}{\boldsymbol{\sigma}_i^2(\mathbf{c}_\theta, t)} \left( \frac{\boldsymbol{\nu}_i'(\mathbf{c}, t) \boldsymbol{\nu}_i(\mathbf{c}_\theta, t)}{\boldsymbol{\nu}_i(\mathbf{c}, t) \boldsymbol{\nu}_i'(\mathbf{c}_\theta, t)} \right) = 1 \tag{53}$$

In the vector form the above equation can be written as,

$$\frac{\boldsymbol{\sigma}_t^2(\mathbf{c}) \boldsymbol{\nu}_t(\mathbf{c}_\theta) \nabla_t \boldsymbol{\nu}(\mathbf{c}, t)}{\boldsymbol{\sigma}_t^2(\mathbf{c}_\theta) \boldsymbol{\nu}_t(\mathbf{c}) \nabla_t \boldsymbol{\nu}(\mathbf{c}_\theta, t)} \to \mathbf{1_d} \tag{54}$$

Eq. 54 holds if:

- $x_\theta = x_0$ i.e. the unet can perfectly map $\mathbf{x}_t$ to $\mathbf{x}_0 \ \forall t \in [0, 1]$ which is unrealistic.

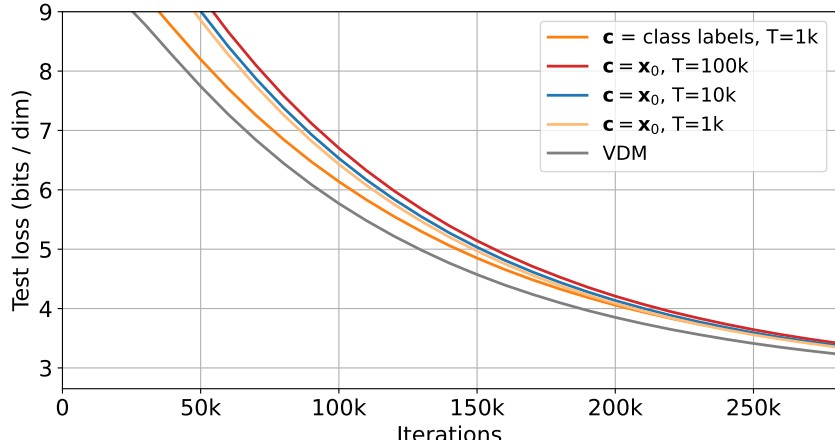

Figure 4: For $\mathbf{c}$ = "class labels" or $\mathbf{c} = \mathbf{x}_0$ the likelihood estimates are worse than VDM. For $\mathbf{c} = \mathbf{x}_0$ we see that the VLB degrades with increasing T whereas for VDM and MULAN it improves for increasing T; see Kingma et al. (2021). This empirical observation is consistent with our mathematical insights earlier. As these models consistently exhibit inferior performance w.r.t VDM, in line with our initial conjectures, we refrain from training them beyond 300k iterations due to the substantial computational cost involved.

- Clever parameterizations for $\boldsymbol{\sigma}, \boldsymbol{\alpha}, \boldsymbol{\nu}$ that ensure Eq. 54 holds.

Because of aforementioned challenges we evaluate this method with finite $T = 1000$. We demonstrate the performance of the model empirically in Fig. 4.

### C.2.2 Recovering VDM

If we substitute $\boldsymbol{\nu}_t(\mathbf{c}), \boldsymbol{\nu}_t(\mathbf{c}_\theta)$ with $\boldsymbol{\nu}(t)$ (since the SNR isn't conditioned on the context $\mathbf{c}$), $\boldsymbol{\sigma}_t(\mathbf{c}_\theta), \boldsymbol{\sigma}_t(\mathbf{c})$ with $\sigma_t$ and $\boldsymbol{\alpha}_t(\mathbf{c}_\theta), \boldsymbol{\alpha}_t(\mathbf{c})$ with $\alpha_t$, Eq. 41 reduces to the intermediate loss in VDM i.e. $\frac{1}{2}(\mathbf{x}_\theta - \mathbf{x}_0)^\top (\nabla_t \boldsymbol{\nu}(t)) (\mathbf{x}_\theta - \mathbf{x}_0)$ and Eq. 51 reduces to 0.

### C.3 Experimental results

In Fig. 4 we demonstrate that the multivariate diffusion processes where $\mathbf{c}$ = "class labels" or $\mathbf{c} = \mathbf{x}_0$ perform worse than VDM. Since a continuous time formulation i.e. $T \to \infty$ for the case when $\mathbf{c} = \mathbf{x}_0$ isn't possible (unlike MULAN or VDM) we evaluate these models in the discrete time setting where we use $T = 1000$. Furthermore we also ablate $T = 10k, 100k$ for $\mathbf{c} = \mathbf{x}_0$ to show that the VLB degrades with increasing T whereas for VDM and MULAN it improves for increasing T; see Kingma et al. (2021). This empirical observation is consistent with our mathematical insights earlier. As these models consistently exhibit inferior performance w.r.t VDM, in line with our initial conjectures, we refrain from training them beyond 300k iterations due to the substantial computational cost involved.

## D MULAN: MUltivariate Latent Auxiliary variable Noise Schedule

### D.1 Noise Reparameterization

Since the forward pass is given by $\mathbf{x}_t = \boldsymbol{\alpha}_t(\mathbf{z})\mathbf{x}_0 + \boldsymbol{\sigma}_t(\mathbf{z})\epsilon_t$ we can write the noise $\epsilon_t$ in terms of $\mathbf{x}_0, \mathbf{x}_t$ in the following manner:

$$\epsilon_t = \frac{\mathbf{x}_t - \alpha_t(\mathbf{z})\mathbf{x}_0}{\sigma_t(\mathbf{z})} \qquad (55)$$

Like Dhariwal & Nichol (2021); Kingma et al. (2021), we parameterize the denoising model in terms of a noise prediction model $\epsilon_\theta(\mathbf{x}_t, \mathbf{z}, t)$,

$$\epsilon_\theta(\mathbf{x}_t, \mathbf{z}, t) = \boldsymbol{\sigma}_t^{-1}(\mathbf{z}) \left( \mathbf{x}_t - \boldsymbol{\alpha}_t(\mathbf{z})\mathbf{x}_\theta(\mathbf{x}_t, \mathbf{z}, t) \right) \tag{56}$$

## D.2 Polynomial Noise Schedule

**Reasons for choosing a degree 5 polynomial.** Let's say a scalar-valued polynomial, $f(x)$, we require it to be monotonically increasing, i.e. $\frac{d}{dx}f(x) \geq 0$ for $x \in \mathbb{R}$, and for some combination of its coefficients have its derivative, $\frac{d}{dx}f(x) = 0$ at least twice in $x \in \mathbb{R}$. For $\frac{d}{dx}f(x) = 0$ be at least twice, its degree has to be at least 2. To ensure $\frac{d}{dx}f(x) \geq 0$, we parameterize $\frac{d}{dx}f(x)$ in the following manner: $\frac{d}{dx}f(x) = (ax^2 + bx + c)^2$. This way $\frac{d}{dx}f(x)$ is always positive and can assume a value of 0 twice for some combination of $(a, b, c) \in \mathbb{R}^3$. Thus $f(x)$ takes the following functional form:

$$\begin{aligned} f(x) &= \int (ax^2 + bx + c)^2 dx \\ &= \frac{a^2}{5}x^5 + \frac{ab}{2}x^4 + \frac{b^2 + 2ac}{3}x^3 + bcx^2 + c^2x + \text{constant}. \end{aligned} \tag{57}$$

$\boldsymbol{\gamma}(\mathbf{c}, t)$ **as a degree 5 polynomial.** For the above-mentioned reasons we $\boldsymbol{\gamma}(\mathbf{c}, t)$ as a degree 5 polynomial in $t$.

$$\gamma_\phi(\mathbf{c}, t) = \gamma_{\min} + (\gamma_{\max} - \gamma_{\min}) \left( \frac{\frac{\mathbf{a}^2(\mathbf{c})}{5}t^5 + \frac{\mathbf{a}(\mathbf{c})\mathbf{b}(\mathbf{c})}{2}t^4 + \frac{\mathbf{b}(\mathbf{c})^2 + 2\mathbf{a}(\mathbf{c})\mathbf{d}(\mathbf{c})}{3}t^3 + \mathbf{b}(\mathbf{c})\mathbf{d}(\mathbf{c})t^2 + \mathbf{d}^2(\mathbf{c})t}{\frac{\mathbf{a}^2(\mathbf{c})}{5} + \frac{\mathbf{a}(\mathbf{c})\mathbf{b}(\mathbf{c})}{2} + \frac{\mathbf{b}(\mathbf{c})^2 + 2\mathbf{a}(\mathbf{c})\mathbf{d}(\mathbf{c})}{3} + \mathbf{b}(\mathbf{c})\mathbf{d}(\mathbf{c}) + \mathbf{d}^2(\mathbf{c})} \right) \tag{58}$$

Notice that $\gamma_\phi(\mathbf{c}, t)$ has these interesting properties:

- Is an increasing function in $t \in \mathbb{R}$ which is crucial as mentioned in Sec. 3.5.
- $\gamma_\phi(\mathbf{c}, t = 0) = \gamma_{\min}$ and $\gamma_\phi(\mathbf{c}, t = 1) = \gamma_{\max}$ is ideal; see (Zheng et al., 2023).
- Its time-derivative i.e. $\nabla_t\gamma_\phi(\mathbf{c}, t)$ can be zero twice in $t \in [0, 1]$. This isn't a necessary condition but it's nice to have a flexible noise schedule whose slope w.r.t $t$ can be 0 at the beginning and the end of the diffusion process.

and its time derivative, $\nabla_t\gamma_\phi(\mathbf{c}, t)$ can assume a value of 0 twice:

Thus $\gamma_\phi(\mathbf{c}, t)$ can be written as:

$$\gamma_\phi(\mathbf{c}, t) = \gamma_{\min} + (\gamma_{\max} - \gamma_{\min}) \left( \frac{\tilde{\mathbf{a}}(\mathbf{c})t^5 + \tilde{\mathbf{b}}(\mathbf{c})t^4 + \tilde{\mathbf{d}}(\mathbf{c})t^3 + \tilde{\mathbf{e}}(\mathbf{c})t^2 + \tilde{\mathbf{f}}(\mathbf{c})t}{\tilde{\mathbf{a}}(\mathbf{c}) + \tilde{\mathbf{b}}(\mathbf{c}) + \tilde{\mathbf{d}}(\mathbf{c}) + \mathbf{e}'(\mathbf{c}) + \mathbf{f}'(\mathbf{c})} \right)$$

where

$$\tilde{\mathbf{a}}(\mathbf{c}) \equiv \frac{\mathbf{a}^2(\mathbf{c})}{5}$$

$$\tilde{\mathbf{b}}(\mathbf{c}) \equiv \frac{\mathbf{a}(\mathbf{c})\mathbf{b}(\mathbf{c})}{2}$$

$$\tilde{\mathbf{d}}(\mathbf{c}) \equiv \frac{\mathbf{b}(\mathbf{c})^2 + 2\mathbf{a}(\mathbf{c})\mathbf{d}(\mathbf{c})}{3}$$

$$\tilde{\mathbf{e}}(\mathbf{c}) \equiv \mathbf{b}(\mathbf{c})\mathbf{d}(\mathbf{c})$$

$$\tilde{\mathbf{f}}(\mathbf{c}) \equiv \mathbf{d}^2(\mathbf{c})$$

## D.3 Sum of Gamma Distribution

Niepert et al. (2021) show that $\mathbf{z} \sim p_\theta(\mathbf{z}; \theta)$ is equivalent to $\mathbf{z} = \arg\max_{y \in Y} \langle \theta + \epsilon_g, y \rangle$ where $\epsilon_g$ is a sample from Sum-of-Gamma distribution given by

$$\text{SoG}(k, \tau, s) = \frac{\tau}{k} \left( \sum_{i=1}^s \text{Gamma}\left(\frac{1}{k}, \frac{k}{i}\right) - \log s \right), \tag{59}$$

where $s$ is a positive integer and $\text{Gamma}(\alpha, \beta)$ is the Gamma distribution with $(\alpha, \beta)$ as the shape and scale parameters.

## D.4  VARIATIONAL LOWER BOUND

In this section we derive the VLB. For ease of reading we use the notation $\mathbf{x}_t$ to denote $\mathbf{x}_{t(i)}$ and $\mathbf{x}_{t-1}$ to denote $\mathbf{x}_{t(i-1)} \equiv \mathbf{x}_{s(i)}$ in the following derivation.

$$
\begin{aligned}
&- \log p_\theta(\mathbf{x}_0) \\
&\leq \mathbb{E}_{q_\phi}\left[-\log \frac{p_\theta(\mathbf{z}, \mathbf{x}_{0:T})}{q_\phi(\mathbf{z}, \mathbf{x}_{1:T}|\mathbf{x}_0)}\right] \\
&= \mathbb{E}_{q_\phi}\left[-\log \frac{p_\theta(\mathbf{x}_{0:T-1}|\mathbf{z}, \mathbf{x}_T)}{q_\phi(\mathbf{z}, \mathbf{x}_{1:T}|\mathbf{x}_0)} - \log p_\theta(\mathbf{x}_T) - \log p_\theta(\mathbf{z})\right] \\
&= \mathbb{E}_{q_\phi}\left[-\log \frac{p_\theta(\mathbf{x}_{0:T-1}|\mathbf{z}, \mathbf{x}_T)}{q_\phi(\mathbf{x}_{1:T}|\mathbf{z}, \mathbf{x}_0)} - \log \frac{1}{q_\phi(\mathbf{z}|\mathbf{x}_0)} - \log p_\theta(\mathbf{x}_T) - \log p_\theta(\mathbf{z})\right] \\
&= \mathbb{E}_{q_\phi}\left[-\log \frac{p_\theta(\mathbf{x}_{0:T-1}|\mathbf{z}, \mathbf{x}_T)}{q_\phi(\mathbf{x}_{1:T}|\mathbf{z}, \mathbf{x}_0)} - \log p_\theta(\mathbf{x}_T) - \log \frac{p_\theta(\mathbf{z})}{q_\phi(\mathbf{z}|\mathbf{x}_0)}\right] \\
&= \mathbb{E}_{q_\phi}\left[-\sum_{t=1}^T \log \frac{p_\theta(\mathbf{x}_{t-1}|\mathbf{z}, \mathbf{x}_t)}{q_\phi(\mathbf{x}_t|\mathbf{x}_{t-1}, \mathbf{z}, \mathbf{x}_0)} - \log p_\theta(\mathbf{x}_T) - \log \frac{p_\theta(z)}{q_\phi(z|\mathbf{x}_0)}\right] \\
&= \mathbb{E}_{q_\phi}\left[-\log \frac{p_\theta(\mathbf{x}_0|\mathbf{z}, \mathbf{x}_1)}{q_\phi(\mathbf{x}_1|\mathbf{x}_0, \mathbf{z})} - \sum_{t=2}^T \log \frac{p_\theta(\mathbf{x}_{t-1}|\mathbf{z}, \mathbf{x}_t)}{q_\phi(\mathbf{x}_t|\mathbf{x}_{t-1}, \mathbf{z}, \mathbf{x}_0)} - \log p_\theta(\mathbf{x}_T) - \log \frac{p_\theta(z)}{q_\phi(\mathbf{z}|\mathbf{x}_0)}\right] \\
&= \mathbb{E}_{q_\phi}\left[-\log \frac{p_\theta(\mathbf{x}_0|\mathbf{z}, \mathbf{x}_1)}{q_\phi(\mathbf{x}_1|\mathbf{x}_0, \mathbf{z})} - \sum_{t=2}^T \log \frac{p_\theta(\mathbf{x}_{t-1}|\mathbf{z}, \mathbf{x}_t)q_\phi(\mathbf{x}_{t-1}|\mathbf{z}, \mathbf{x}_0)}{q_\phi(\mathbf{x}_{t-1}|\mathbf{x}_t, \mathbf{z}, \mathbf{x}_0)q_\phi(\mathbf{x}_t|\mathbf{z}, \mathbf{x}_0)} - \log p_\theta(\mathbf{x}_T) - \log \frac{p_\theta(\mathbf{z})}{q_\phi(\mathbf{z}|\mathbf{x}_0)}\right] \\
&= \mathbb{E}_{q_\phi}\left[-\log \frac{p_\theta(\mathbf{x}_0|\mathbf{z}, \mathbf{x}_1)}{q_\phi(\mathbf{x}_1|\mathbf{x}_0, \mathbf{z})} - \sum_{t=2}^T \log \frac{p_\theta(\mathbf{x}_{t-1}|\mathbf{z}, \mathbf{x}_t)}{q_\phi(\mathbf{x}_{t-1}|\mathbf{x}_t, \mathbf{z}, \mathbf{x}_0)} - \sum_{t=2}^T \log \frac{q_\phi(\mathbf{x}_{t-1}|\mathbf{z}, \mathbf{x}_0)}{q_\phi(\mathbf{x}_t|\mathbf{z}, \mathbf{x}_0)} - \log p_\theta(\mathbf{x}_T) - \log \frac{p_\theta(z)}{q_\phi(\mathbf{z}|\mathbf{x}_0)}\right] \\
&= \mathbb{E}_{q_\phi}\left[-\log \frac{p_\theta(\mathbf{x}_0|\mathbf{z}, \mathbf{x}_1)}{q_\phi(\mathbf{x}_1|\mathbf{x}_0, \mathbf{z})} - \sum_{t=2}^T \log \frac{p_\theta(\mathbf{x}_{t-1}|\mathbf{z}, \mathbf{x}_t)}{q_\phi(\mathbf{x}_{t-1}|\mathbf{x}_t, \mathbf{z}, \mathbf{x}_0)} - \log \frac{q(\mathbf{x}_1|\mathbf{z}, x_0)}{q_\phi(\mathbf{x}_T|\mathbf{z}, \mathbf{x}_0)} - \log p_\theta(\mathbf{x}_T) - \log \frac{p_\theta(\mathbf{z})}{q_\phi(\mathbf{z}|\mathbf{x}_0)}\right] \\
&= \mathbb{E}_{q_\phi}\left[-\log p_\theta(\mathbf{x}_0|\mathbf{z}, \mathbf{x}_1) - \sum_{t=2}^T \log \frac{p_\theta(\mathbf{x}_{t-1}|\mathbf{z}, \mathbf{x}_t)}{q_\phi(\mathbf{x}_{t-1}|\mathbf{x}_t, \mathbf{z}, \mathbf{x}_0)} - \log \frac{1}{q_\phi(\mathbf{x}_T|\mathbf{z}, \mathbf{x}_0)} - \log p_\theta(\mathbf{x}_T) - \log \frac{p_\theta(\mathbf{z})}{q_\phi(\mathbf{z}|\mathbf{x}_0)}\right] \\
&= \mathbb{E}_{q_\phi}\left[-\log p_\theta(\mathbf{x}_0|\mathbf{z}, \mathbf{x}_1) - \sum_{t=2}^T \log \frac{p_\theta(\mathbf{x}_{t-1}|z, \mathbf{x}_t)}{q_\phi(\mathbf{x}_{t-1}|\mathbf{x}_t, \mathbf{z}, \mathbf{x}_0)} - \log \frac{p_\theta(\mathbf{x}_T)}{q_\phi(\mathbf{x}_T|\mathbf{z}, \mathbf{x}_0)} - \log \frac{p_\theta(\mathbf{z})}{q_\phi(z|\mathbf{x}_0)}\right] \\[2ex]
&= \mathbb{E}_{q_\phi}\left[\underbrace{-\log p_\theta(\mathbf{x}_0|\mathbf{z}, \mathbf{x}_1)}_{\mathcal{L}_{\text{recons}}} + \underbrace{\sum_{t=2}^T D_{\text{KL}}\big[p_\theta(\mathbf{x}_{t-1}|\mathbf{z}, \mathbf{x}_t)\|q_\phi(\mathbf{x}_{t-1}|\mathbf{x}_t, \mathbf{z}, \mathbf{x}_0)\big]}_{\mathcal{L}_{\text{diffusion}}}\right] \\[2ex]
&\quad + \mathbb{E}_{q_\phi}\left[\underbrace{D_{\text{KL}}\big[p_\theta(\mathbf{x}_T)\|q_\phi(\mathbf{x}_T|\mathbf{z}, \mathbf{x}_0)\big]}_{\mathcal{L}_{\text{prior}}} + \underbrace{D_{\text{KL}}\big[p_\theta(\mathbf{z})\|q(\mathbf{z}|\mathbf{x}_0)\big]}_{\mathcal{L}_{\text{latent}}}\right] \qquad (60)
\end{aligned}
$$

Switching back to the notation used throughout the paper, the VLB is given as:

$$
\begin{aligned}
& -\log p_\theta(\mathbf{x}_0) \\
&= \mathbb{E}_{q_\phi} \Bigg[ \underbrace{-\log p_\theta(\mathbf{x}_0|\mathbf{z}, \mathbf{x}_1)}_{\mathcal{L}_{\text{recons}}} + \underbrace{\sum_{i=2}^{T} \mathrm{D}_{\mathrm{KL}}[p_\theta(\mathbf{x}_{s(i)}|\mathbf{z}, \mathbf{x}_{t(i)}) \| q_\phi(\mathbf{x}_{s(i)}|\mathbf{x}_{t(i)}, \mathbf{z}, \mathbf{x}_0)]}_{\mathcal{L}_{\text{diffusion}}} \Bigg] \\
&\quad + \mathbb{E}_{q_\phi} \Bigg[ \underbrace{\mathrm{D}_{\mathrm{KL}}[p_\theta(\mathbf{x}_1) \| q_\phi(\mathbf{x}_T|\mathbf{z}, \mathbf{x}_0)]}_{\mathcal{L}_{\text{prior}}} + \underbrace{\mathrm{D}_{\mathrm{KL}}[p_\theta(\mathbf{z}) \| q(\mathbf{z}|\mathbf{x}_0)]}_{\mathcal{L}_{\text{latent}}} \Bigg]
\end{aligned}
\tag{61}
$$

## D.5 DIFFUSION LOSS

To derive the diffusion loss, $\mathcal{L}_{\text{diffusion}}$ in Eq. 9, we first derive an expression for $\mathrm{D}_{\mathrm{KL}}(q_\phi(\mathbf{x}_s|\mathbf{z}, \mathbf{x}_t, \mathbf{x}_0) \| p_\theta(\mathbf{x}_s|\mathbf{z}, \mathbf{x}_t))$ using Eq. 4 and Eq. 6 in the following manner (details in Suppl. D):

$$
\begin{aligned}
& \mathrm{D}_{\mathrm{KL}}(q_\phi(\mathbf{x}_s|\mathbf{z}, \mathbf{x}_t, \mathbf{x}_0) \| p_\theta(\mathbf{x}_s|\mathbf{z}, \mathbf{x}_t)) \\
&= \frac{1}{2} \left( (\boldsymbol{\mu}_{q_\phi} - \boldsymbol{\mu}_p)^\top \boldsymbol{\Sigma}_\theta^{-1} (\boldsymbol{\mu}_{q_\phi} - \boldsymbol{\mu}_p) + \mathrm{tr}\left( \boldsymbol{\Sigma}_{q_\phi} \boldsymbol{\Sigma}_p^{-1} - \mathbf{I}_n \right) - \log \frac{|\boldsymbol{\Sigma}_{q_\phi}|}{|\boldsymbol{\Sigma}_p|} \right) \\
&= \frac{1}{2} \left( (\mathbf{x}_0 - \mathbf{x}_\theta)^\top \mathrm{diag}(\boldsymbol{\nu}(\mathbf{z}, s) - \boldsymbol{\nu}(\mathbf{z}, t))(\mathbf{x}_0 - \mathbf{x}_\theta) \right)
\end{aligned}
\tag{62}
$$

Let $\lim_{T \to \infty} T(\boldsymbol{\nu}_s(z) - \boldsymbol{\nu}_t(z)) = -\nabla_t \boldsymbol{\nu}(\mathbf{z}, t)$ be the partial derivative of the vector $\boldsymbol{\nu}(\mathbf{z}, t)$ w.r.t scalar $t$. Then we derive the diffusion loss, $\mathcal{L}_{\text{diffusion}}$, for the continuous case in the following manner

(for brevity we use the notation $s$ for $s(i) = (i-1)/T$ and $t$ for $t(i) = i/T$):

$\mathcal{L}_{\text{diffusion}}$

$$= \lim_{T \to \infty} \frac{1}{2} \sum_{i=2}^{T} \mathbb{E}_{\epsilon \sim \mathcal{N}(0, \mathbf{I}_n)} D_{\text{KL}}(q(\mathbf{x}_s | \mathbf{x}_t, \mathbf{x}_0, \mathbf{z}) \| p_\theta(\mathbf{x}_s | \mathbf{x}_t, \mathbf{z}))$$

Using Eq. 62 we get,

$$= \lim_{T \to \infty} \frac{1}{2} \sum_{i=2}^{T} \mathbb{E}_{\epsilon \sim \mathcal{N}(0, \mathbf{I}_n)} (\mathbf{x}_0 - \mathbf{x}_\theta(\mathbf{x}_t, t(i)))^\top \text{diag}\left(\boldsymbol{\nu}(s(i), \mathbf{z}) - \boldsymbol{\nu}(t(i), \mathbf{z})\right) (\mathbf{x}_0 - \mathbf{x}_\theta(\mathbf{x}_t, t(i)))$$

$$= \frac{1}{2} \mathbb{E}_{\epsilon \sim \mathcal{N}(0, \mathbf{I}_n)} \left[ \lim_{T \to \infty} \sum_{i=2}^{T} T(\mathbf{x}_0 - \mathbf{x}_\theta(\mathbf{x}_t, t(i)))^\top \text{diag}\left(\boldsymbol{\nu}(s(i), \mathbf{z}) - \boldsymbol{\nu}(t(i), \mathbf{z})\right) (\mathbf{x}_0 - \mathbf{x}_\theta(\mathbf{x}_t, t(i))) \frac{1}{T} \right]$$

Using the fact that $\lim_{T \to \infty} T\left(\boldsymbol{\nu}(s, \mathbf{z}) - \boldsymbol{\nu}(\mathbf{z}, t)\right) = -\nabla_t \boldsymbol{\nu}(t, \mathbf{z})$ we get,

$$= -\frac{1}{2} \mathbb{E}_{t \sim \{0,\ldots,1\}} \left[ (\mathbf{x}_0 - \mathbf{x}_\theta(\mathbf{x}_t, t))^\top (\nabla_t \boldsymbol{\nu}_t(z)) (\mathbf{x}_0 - \mathbf{x}_\theta(\mathbf{x}_t, t)) \right]$$

Substituting $\mathbf{x}_0 = \boldsymbol{\alpha}_t^{-1}(\mathbf{z})(\mathbf{x}_t - \boldsymbol{\sigma}_t(\mathbf{z})\epsilon_t)$ from Eq. 55 and

Substituting $\mathbf{x}_\theta(\mathbf{x}_t, \mathbf{z}; t) = \boldsymbol{\alpha}_t^{-1}(\mathbf{z})(\mathbf{x}_t - \boldsymbol{\sigma}_t(\mathbf{z})\epsilon_\theta(\mathbf{x}_t, t))$ from Eq. 56 we get,

$$= -\frac{1}{2} \mathbb{E}_{t \sim [0,1]} \left[ (\epsilon_t - \epsilon_\theta(\mathbf{x}_t, t))^\top \left( \frac{\boldsymbol{\sigma}_t^2(\mathbf{z})}{\boldsymbol{\alpha}_t^2(\mathbf{z})} \times \nabla_t \boldsymbol{\nu}_t(\mathbf{z}) \right) (\epsilon_t - \epsilon_\theta(\mathbf{x}_t, t)) \right]$$

Let $\boldsymbol{\nu}^{-1}(\mathbf{z}, t)$ denote the reciprocal of the values in the vector $\boldsymbol{\nu}(\mathbf{z}, t)$.

$$= -\frac{1}{2} \mathbb{E}_{t \sim [0,1]} \left[ (\epsilon_t - \epsilon_\theta(\mathbf{x}_t, t))^\top \text{diag}\left(\boldsymbol{\nu}^{-1}(t)(\mathbf{z})\nabla_t \boldsymbol{\nu}_t(\mathbf{z})\right) (\epsilon_t - \epsilon_\theta(\mathbf{x}_t, t)) \right]$$

Substituting $\boldsymbol{\nu}(\mathbf{z}, t) = \exp(-\boldsymbol{\gamma}(\mathbf{z}, t))$ from Sec. D.1

$$= -\frac{1}{2} \mathbb{E}_{t \sim [0,1]} \left[ (\epsilon_t - \epsilon_\theta(\mathbf{x}_t, t))^\top \text{diag}\left(\exp\left(\boldsymbol{\gamma}(\mathbf{z}, t)\right) \nabla_t \exp\left(-\boldsymbol{\gamma}(\mathbf{z}, t)\right)\right) (\epsilon_t - \epsilon_\theta(\mathbf{x}_t, t)) \right]$$

$$= \frac{1}{2} \mathbb{E}_{t \sim [0,1]} \left[ (\epsilon_t - \epsilon_\theta(\mathbf{x}_t, t))^\top \text{diag}\left(\exp\left(\boldsymbol{\gamma}(\mathbf{z}, t)\right) \exp\left(-\boldsymbol{\gamma}(\mathbf{z}, t)\right) \nabla_t \boldsymbol{\gamma}(\mathbf{z}, t)\right) (\epsilon_t - \epsilon_\theta(\mathbf{x}_t, t)) \right]$$

$$= \frac{1}{2} \mathbb{E}_{t \sim [0,1]} \left[ (\epsilon_t - \epsilon_\theta(\mathbf{x}_t, t))^\top \text{diag}\left(\nabla_t \boldsymbol{\gamma}(\mathbf{z}, t)\right) (\epsilon_t - \epsilon_\theta(\mathbf{x}_t, t)) \right] \tag{63}$$

## D.6 Recovering VDM from the Vectorized Representation of the diffusion loss

Notice that we recover the loss function in VDM when $\boldsymbol{\nu}(\mathbf{z}, t) = \nu(t)\mathbf{1_d}$ where $\nu_t \in \mathbb{R}^+$ and $\mathbf{1_d}$ represents a vector of 1s of size $d$ and the noising schedule isn't conditioned on $\mathbf{z}$.

$$\int_0^1 \langle \mathbf{f}_\theta(\mathbf{x}_0, \boldsymbol{\nu}(\mathbf{z}, t)), \frac{d}{dt}\boldsymbol{\nu}(t)\rangle dt = \int_0^1 \langle \mathbf{f}_\theta(\mathbf{x}_0, \boldsymbol{\nu}(t)), \frac{d}{dt}(\nu(t)\mathbf{1_n})\rangle dt$$

$$= \int_0^1 \langle \mathbf{f}_\theta(\mathbf{x}_0, \boldsymbol{\nu}(t)), \mathbf{1_d}\rangle \frac{d}{dt}\nu(t) dt$$

$$= \int_0^1 \frac{d}{dt}\nu(t)\|\mathbf{f}_\theta(\mathbf{x}_0, \boldsymbol{\nu}(t))\|_1^1 dt$$

$$= \int_0^1 \frac{d}{dt}\nu(t)\|(\mathbf{x}_0 - \tilde{\mathbf{x}}_\theta(\mathbf{x}_{\boldsymbol{\nu}(t)}, \boldsymbol{\nu}(t)))\|_2^2 dt \tag{64}$$

$\int_0^1 \frac{d}{dt}\nu(t)\|(\mathbf{x}_0 - \tilde{\mathbf{x}}_\theta(\mathbf{x}_{\boldsymbol{\nu}(t)}, \boldsymbol{\nu}(t)))\|_2^2 dt$ denotes the diffusion loss, $\mathcal{L}_{\text{diffusion}}$, as used in VDM; see Kingma et al. (2021).

## E Subset Sampling

Sampling a subset of $k$ items from a collection of collection of $n$ items, $x_1, x_2, \ldots, x_3$ belongs to a category of algorithms called reservoir algorithms [TODO]. In weighted reservoir sampling,

every $x_i$ is associated with a weight $w_i \geq 0$. The probability associated with choosing the sequence $S_{\mathrm{wrs}} = [i_1, i_2, \ldots, i_k]$ be a tuple of indices. Then the probability associated with sampling this sequence is

$$p(S_{\mathrm{wrs}}|\mathbf{w}) = \frac{w_{i_1}}{Z} \frac{w_{i_2}}{Z - w_{i_1}} \cdots \frac{w_{i_k}}{Z - \sum_{j=1}^{k-1} w_{i_j}} \tag{65}$$

Efraimidis & Spirakis (2006) give an algorithm for weighted reservoir sampling where each item is assigned a random key $r_i = u_i^{\frac{1}{w_i}}$ where $u_i$ is drawn from a uniform distribution [0, 1] and $w_i$ is the weight of item $x_i$. Let TopK$(\mathbf{r}, k)$ which takes keys $\mathbf{r} = [r_1, r_2, \ldots, r_n]$ and returns a sequence $[i_1, i_2, \ldots, i_k]$. Efraimidis & Spirakis (2006) proved that TopK$(\mathbf{r}, k)$ is distributed according to $p(S_{\mathrm{wrs}}|\mathbf{w})$.

Let's represent a subset $S \in \{0, 1\}^n$ with exactly $k$ non-zero elements that are equal to 1. Then the probability associated with sampling $S$ is given as,

$$p(S|\mathbf{w}) = \sum_{S_{\mathrm{wrs}} \in \Pi(S)} p(S_{\mathrm{wrs}}|\mathbf{w}) \tag{66}$$

where $\Pi(S)$ denotes all possible permutations of the sequence $S$. By ignoring the ordering of the elements in $S_{\mathrm{wrs}}$ we can sample using the same algorithm. Xie & Ermon (2019) show that this sampling algorithm is equivalent to TopK$(\hat{\mathbf{r}}, k)$ where $\hat{\mathbf{r}} = [\hat{r}_1, \hat{r}_2, \ldots, \hat{r}_n]$ where $\hat{r}_i = -\log(-\log(r_i)) = \log w_i + $ Gumbel(0, 1). This holds true because the monotonic transformation $-\log(-\log(x))$ preserves the ordering of the keys and thus TopK$(\mathbf{r}, k) \equiv$ TopK$(\hat{\mathbf{r}}, k)$.

Niepert et al. (2021) show that adding SOG noise instead of Gumbel noise leads to better performance.

And hence, given logits $\log \mathbf{w}$, we sample a $k$-hot vector using TopK$(\log \mathbf{w} + \epsilon)$. We choose a categorical prior with uniform distribution across $n$ classes. Thus the KL loss term is given by:

$$-\sum_{i=1}^{n} \frac{w_i}{Z} \log \left( n \frac{w_i}{Z} \right) \tag{67}$$

## F   MODEL ARCHITECTURE

Our model architecture is extremely similar to VDM. The UNet of our pixel-space diffusion has an unchanged architecture from Kingma et al. (2021). We change the noise scheduling from a learned linear one to a fifth degree polynomial as seen in section 3.2.2. We also use a UNet of the per pixel noising schedule which has an identical architecture to the diffusion models' UNet while the encoder is just an MLP. We employ a residual architecture that first learns a scalar global value and then has a network the predicts a per-pixel delta to that value.

# G VISUALIZATIONS

## G.1 CIFAR-10

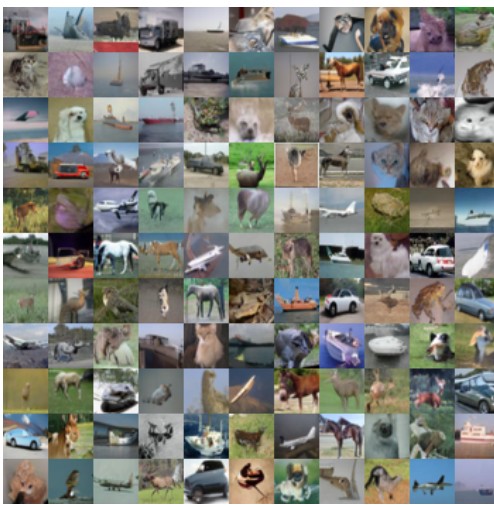 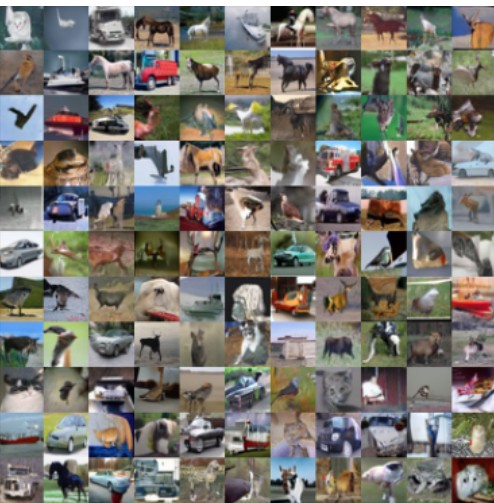

(a) VDM after 10M training iterations.

(b) MULAN with velocity reparameterization after 8M training iterations.

Figure 5: CIFAR-10 samples generated by different methods.

## G.2 IMAGENET

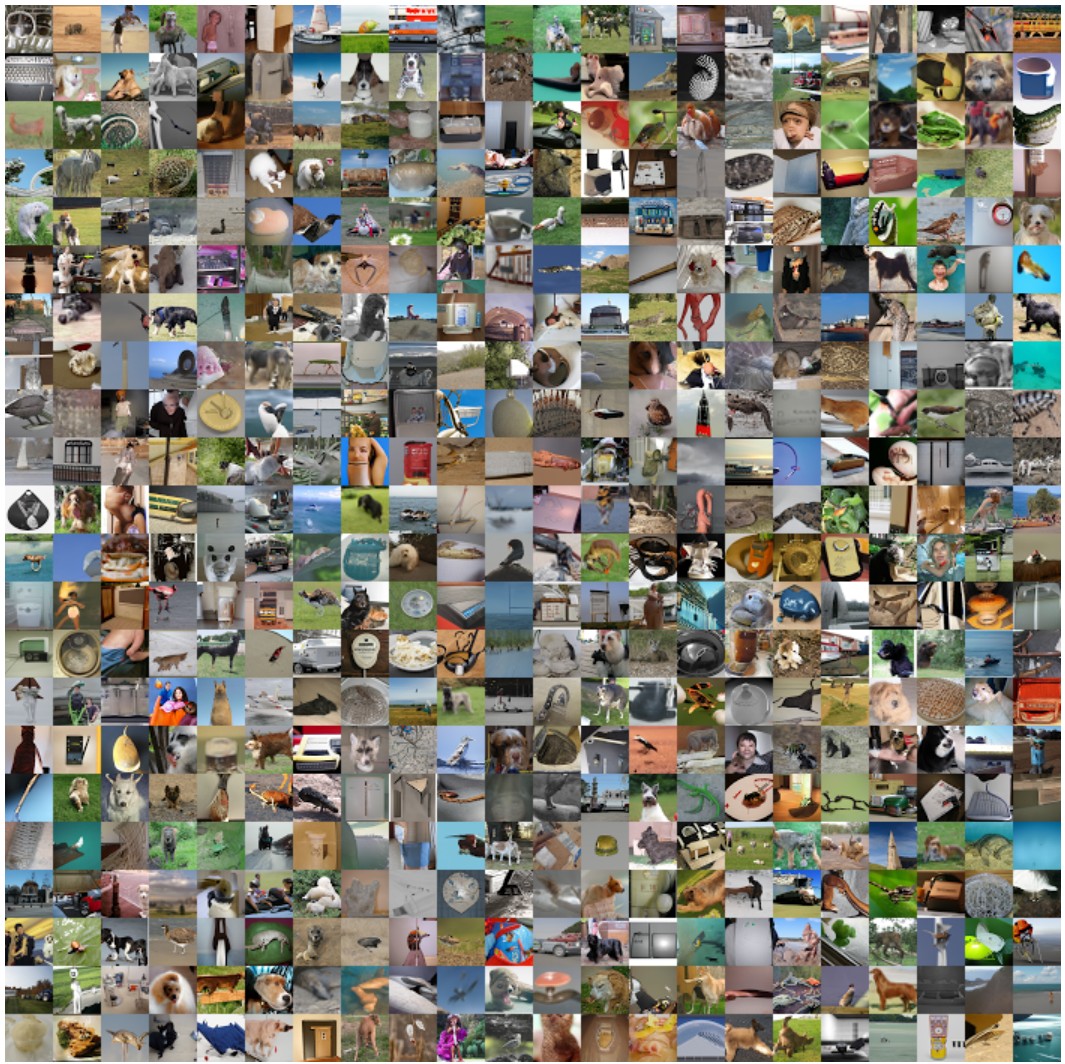

Figure 6: VDM after 2M training iterations.

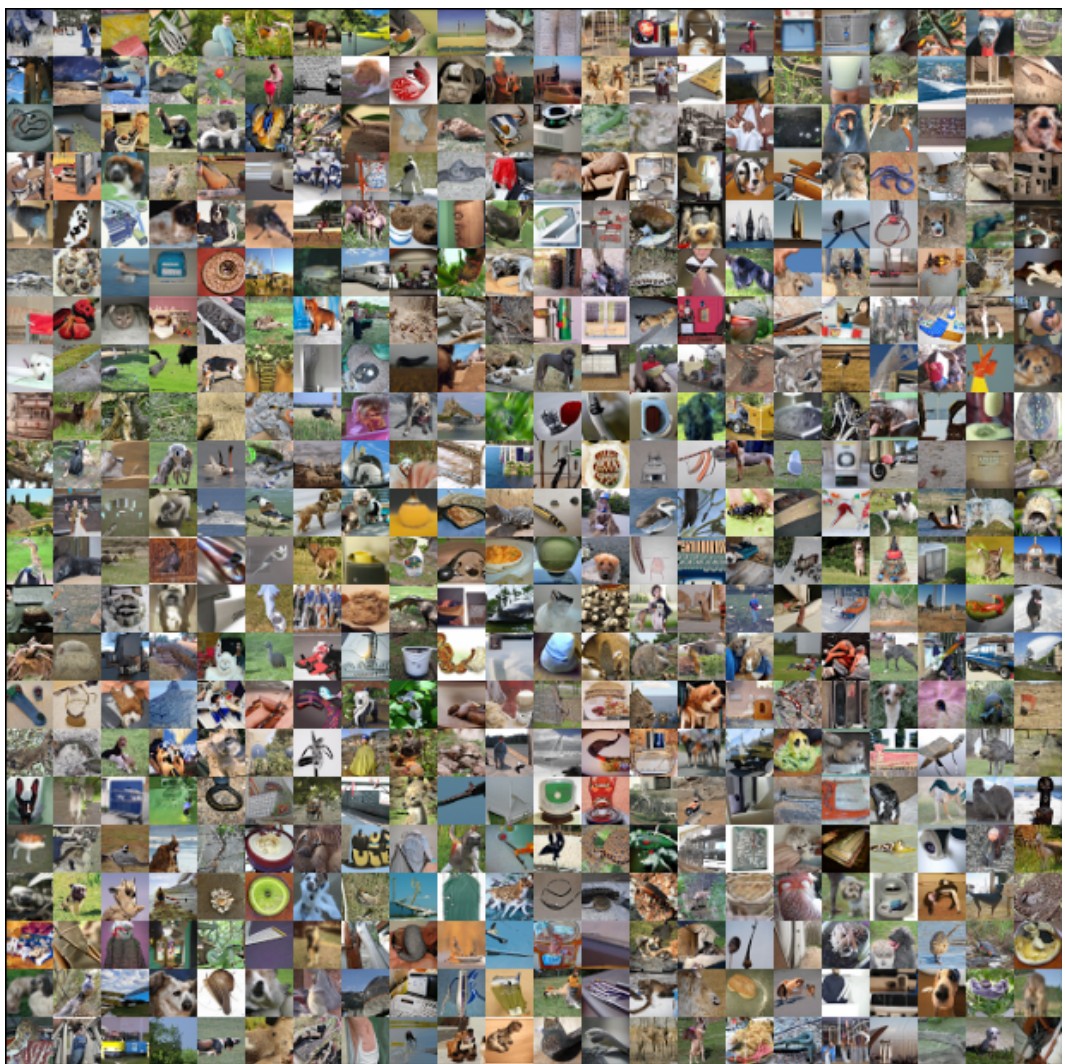

Figure 7: MᴜLAN with $\epsilon$ reparameterization after 2M training iterations.

