# OpenReview forum: "Diffusion Models With Learned Adaptive Noise Processes"
_ICLR.cc/2024/Conference — Submitted to ICLR 2024_

### Official Review · Reviewer_aHz6 · 2023-10-31

**Soundness:** 3 good
**Presentation:** 3 good
**Contribution:** 2 fair
**Rating:** 6
**Confidence:** 3

**Summary:**

In this paper, the authors propose a method to learn the parametric diffusion noise schedule by jointly optimizing model parameters and diffusion parameters. In addition, the authors propose a learning method for conditional diffusion via a latent distribution.

**Strengths:**

1. The proposed approch that learns an adaptive diffusion noise schedule is somewhat novel.

2.  The paper is well-written and well-organized.

**Weaknesses:**

1.  The authors argue a novel approch that learns conditional diffusion via auxiliary latent variables.
 However,  the relationship and difference compared with (Wang et al. 2023) is not clearly discussed.


2. The advantage of the proposed approch via auxiliary latent variables is not well supported.   In Figure 1 (a), it seems that MuLAN w/o auxiliary latent variable performs worse than the standard VDM.


3. The empirical results can not support the claimed advantage of the proposed method MULAN .  In Table 1, it seems that the proposed method MULAN   performs worse than i-DODE∗ (Zheng et al., 2023).

**Questions:**

Q1.  The empirical results are not convincing enough to demonstrate the advantage of the proposed method.  Could the authors provide additional empirical evidence to support the claim?

Q2.  It seems that the proposed method incurs additional time complexity. Could the authors provide additional running time comparison with baselines?

---

> ### Author Response · Authors · 2023-11-22
> **Response for Reviewer aHz6**
>
> We want to thank the reviewer for their constructive feedback. We address each concern below.
>
> **Concern 1**: The strength of the empirical results. Can the authors provide additional empirical evidence?
>
> We have improved our experimental results, and **we now achieve a new state-of-the-art** in density estimation on CIFAR10 and ImageNet. The table below summarizes our results.
>
> | Method | CIFAR10 bpd | ImageNet bpd |
> |-------------|-------------|-------------|
> | PixelCNN (Van den Oord et al., 2016) | 3.03 | 3.83
> Image Transformer (Parmar et al., 2018) | 2.90 | 3.77
> Score SDE (Song et al., 2020) | 2.99 | -
> Improved DDPM (Nichol & Dhariwal, 2021) | 2.94 | -
> VDM (Kingma et al., 2021) | 2.65 | 3.72
> iDODE (Zheng et al., 2023) | 2.56  | 3.69
> MuLAN (ours; initial version) | 2.60 ± 1e-3 | 3.71 ± 1e-3
> MuLAN (**ours; rebuttal version; state-of-the-art**) | **2.55** ± 1e-3 | **3.67\*** ± 1e-3
>
> *\* Partial result obtained on 70% of the data due to time constraints.*
>
> Our updated method adds two architectural improvements: velocity reparameterization (as in Zheng et al., 2023) and truncated normal dequantization (Dinh et al., 2017; Salimans et al., 2017; Ho et al., 2019; Zheng et al., 2023). The recent iDODE paper contains additional innovations (e.g., importance sampled training) that could further improve the performance of MuLAN. Lastly, for both iDODE and MuLAN with velocity parameterization, we report an importance-sampled estimate of test NLL with K=20 samples, as in the i-DODE paper (sec A.1., eqn. 32).
>
> **Concern 2**: Explaining the relationship between Infodiffusion (Wang et al., 2023) and MuLAN.
>
> In brief, auxiliary-variable diffusion is a technique introduced by Wang et al., 2023. MuLAN makes use of this technique in conjunction with several other components. The resulting method is substantially different from InfoDiffusion and solves a different problem: density estimation versus representation learning.
>
> Specifically, the key novel component of MuLAN is a learned adaptive noise process. A widely held assumption is that the ELBO objective of a diffusion model is invariant to the noise process (Kingma et al., 2021). We **dispel this assumption**: we show that when input-conditioned noise is combined with (a) multivariate noise, (b) a novel polynomial parameterization, and (c) auxiliary variables, a learned noise process yields an improved variational posterior and a tighter ELBO. This approach sets a new state-of-the-art in density estimation.
>
> The table below summarizes the relationship between MuLAN and other methods.
>
>
>
> | | Learned noise | Multivariate noise | Input Conditioned noise | Auxiliary latents | Noise parameterization |
> |---|---|---|---|--- |---|
> VDM (Kingma et al., 2021) | **Yes** | No |No | No | Monotonic neural network |
> Blurring Diffusion Model (Hoogeboom et al., 2022) | No | **Yes** | No | No | Frequency scaling |
> InfoDiffusion (Wang et al., 2023) | No | No | No | In denoising process | Cosine schedule |
> MuLAN (ours) | **Yes** | **Yes** | **Yes** | In nosing & denoising processes | Polynomial, sigmoid
>
> **Concern 3**: MuLAN without auxiliary variables performs worse than VDM.
>
> This may be a misunderstanding. MuLAN without auxiliary variables __performs the same__ as VDM: in Fig 1a, the blue and gray lines overlap almost perfectly. This is expected: when the noise is not adaptive (on the data via auxiliary variables), there is no improvement.
>
> Perhaps the reviewer meant to ask why MuLAN without multivariate noise is worse than VDM? First, recall that when noise is univariate, the ELBO is invariant to the choice of noise process (Kingma et al., 2021). However, auxiliary variables require applying the ELBO twice: this yields a training objective that bounds the marginal log-likelihood less tightly than in VDM. By itself, this lowers performance; however this performance drop is more than compensated by learning the noise process.
>
> **Concern 4**: Understanding the time complexity of the method.
>
> When trained on 8 A100 GPUs, VDM achieves a training rate of 11.6 steps/second, while Mulan trains slightly slower at 10.1 steps/second due to the inclusion of an additional encoder network. However, despite this slower training pace, VDM requires 30 days to reach a BPD of 2.65, whereas Mulan achieves the same BPD within a significantly shorter timeframe of 10 days.

---

### Official Review · Reviewer_U1Ra · 2023-10-31

**Soundness:** 2 fair
**Presentation:** 2 fair
**Contribution:** 2 fair
**Rating:** 5
**Confidence:** 3

**Summary:**

In this work, the authors propose using an instance-dependent multivariate Gaussian noise scheduling and auxiliary latent variables to improve the likelihood estimation. Their method demonstrates strong performance in terms of negative log-likelihood (**NLL**) and convergence.

**Strengths:**

1. Overall, this paper is clearly written and easy to follow. The idea is simple, narrowing the gap between marginal log-likelihood and its evidence lower bound (**ELBO**) by specifying a more flexible family of approximate variational posteriors, which is a standard approach in variational inference. The relevant derivations in the paper are also straightforward.
2. The paper studies the effect of an adaptive multivariate Gaussian noise schedule on the likelihood estimation performance, which can be potentially combined with other techniques to improve the metric. The proposed method, MULAN, is also found advantageous to previous SOTA generative modeling methods in likelihood estimation and convergence. The ablation study underscores the indispensable synergy between the method's core components: the auxiliary variable and the multivariate Gaussian noise scheduling.

**Weaknesses:**

1. The intuition behind using a non-identical pixel-wise Gaussian noise schedule from a frequency perspective (e.g., texture and shape, which are mainly perceptual) is not convincing. It is known that likelihood generally does not correlate with sample quality and visual appearance [1].
2. As mentioned in the paper, the proposed method itself does not contain much novelty. The use of multivariate non-isotropic Gaussian noise scheduling and the introduction of an auxiliary variable in diffusion models are not new [2][3]. The mathematical derivations presented in this paper largely mirror previous work.
3. The introduction of an auxiliary variable does not necessarily agree with the objective of narrowing the posterior gap. In Section 3.3.2, the right-hand side of the first inequality, i.e., Equation (7), to my understanding, is the same as the ELBO of a variational autoencoder. If the actual objective of MULAN is based on the second inequality, then the ELBO w.r.t. $(x, z)$ would act as a bottleneck of the ELBO w.r.t. $(x_0, x_{1..T}, z)$.
4. The experiment results are rather not impressive. As mentioned in the paper, the authors implement their method based on the VDM codebase and adopt the same settings for the most part. VDM is almost the strongest model excluding i-DODE and MULAN (this work) in the main table (Table 1). Although the improvement of the proposed method seems significant compared with other methods, it is far less impressive relative to the result by VDM (the method it is built upon), considering the extra degrees of freedom.
5. Typos:
	Appendix C.1 prexisting -> pre-existing

[1] Theis, Lucas, Aäron van den Oord, and Matthias Bethge. "A note on the evaluation of generative models." arXiv preprint arXiv:1511.01844 (2015).

[2] Hoogeboom, Emiel, and Tim Salimans. "Blurring diffusion models." arXiv preprint arXiv:2209.05557 (2022).

[3] Wang, Yingheng, et al. "InfoDiffusion: Representation Learning Using Information Maximizing Diffusion Models." arXiv preprint arXiv:2306.08757 (2023).

**Questions:**

1. How is BPD calculated? Is it calculated by the stochastic VLB or ODE-based exact likelihood computation methods? If the reported metric of MULAN is obtained by the former one, what is the variance of it? And what effect does the choice of log-SNR parameterization have on the variance? I think the variance of stochastic VLB matters in this case when it is used to compare the proposed method with others including VDM. VDM explicitly minimizes VLB variance with a learned noise schedule whereas MULAN does not.
2. Does the authors try to analyze the auxiliary context variables? Do they have interpretable meaning? If so, it might also be a way to do representation learning and controllable generation. In the paper, the auxiliary variables are also referred to as the context and are said to "encapsulate high-level information".

---

> ### Author Response · Authors · 2023-11-23
> **Response to weaknesses for reviewer U1Ra**
>
> We want to thank the reviewer for their constructive feedback. We address each concern below.
>
> **Concern 1**: The experimental results are not impressive
>
> We report new experimental results, and **we now achieve a new state-of-the-art in density estimation on CIFAR10 and ImageNet**. The table below summarizes our results.
>
> | Method | CIFAR10 bpd | ImageNet bpd |
> | --- | --- | --- |
> |PixelCNN (Van den Oord et al., 2016) | 3.03 | 3.83|
> |Image Transformer (Parmar et al., 2018) | 2.90 | 3.77|
> | Score SDE (Song et al., 2020) | 2.99 | - |
> Improved DDPM (Nichol & Dhariwal, 2021) | 2.94 | -
> | VDM (Kingma et al., 2021) | 2.65 | 3.72
> |i-DODE (Zheng et al., 2023) | 2.56 | 3.69
> MuLAN (ours; initial version) | 2.60 ± 1e-3 | 3.71 ± 1e-3
> MuLAN (**ours; rebuttal version; state-of-the-art**) | **2.55** ± 1e-3 | **3.67\*** ± 1e-3
>
> \* *Partial result obtained on 70% of the data due to time constraints*
>
> Our updated method adds two architectural improvements: velocity reparameterization (as in Zheng et al., 2023) and truncated normal dequantization (Dinh et al., 2017; Salimans et al., 2017; Ho et al., 2019; Zheng et al., 2023). The recent iDODE paper contains additional innovations (e.g., importance sampled training) that could further improve the performance of MuLAN. Lastly, for both iDODE and MuLAN with velocity parameterization, we report an importance-sampled estimate of test NLL with K=20 samples, as in the i-DODE paper (sec A.1., eqn. 32).
>
> Note also that **the magnitude of our improvement over VDM or i-DODE is significant**: it is comparable to the progress made by most papers on these benchmarks from 2018-2021.
>
> **Concern 2**: Understanding the novelty of the paper relative to recent work
>
> MuLAN is the **first** method to introduce a learned adaptive noise process. A widely held assumption is that the ELBO objective is invariant to the noise process (Kingma et al., 2021). We __dispel this assumption__: when input-conditioned noise is combined with (a) multivariate noise, (b) a novel polynomial parameterization, and (c) auxiliary variables, a learned noise process yields a tighter ELBO and a new state-of-the-art in density estimation. While (a), (c) were proposed in other contexts, we leverage them as subcomponents of a novel algorithm.
>
> The table below summarizes the relationship between MuLAN and other methods.
>
> | | Learned noise | Multivariate noise | Input Conditioned noise | Auxiliary latents | Noise parameterization |
> | --- | --- | --- | --- | ---  | --- |
> VDM (Kingma et al., 2021) | **Yes** | No |No | No | Monotonic neural network |
> Blurring Diffusion Model (Hoogeboom et al., 2022) | No | **Yes** | No | No | Frequency scaling |
> InfoDiffusion (Wang et al., 2023) | No | No | No | In denoising process | Cosine schedule |
> MuLAN (ours) | **Yes** | **Yes** | **Yes** | In noising & denoising processes | Polynomial, sigmoid
>
>
>
> **Concern 3**: The motivation of our work, specifically the intuition behind using adaptive noise from a frequency perspective, is not convincing.
>
> The main motivation of our work comes from __variational inference__: we view the noising process as a variational posterior, and we learn it to obtain a tighter ELBO (see Section 3.1). This in turn yields state-of-the-art density estimation.
>
> Note that the intuition to which the reviewer refers is not the motivation for our work. We also do not claim that likelihood correlates with image quality. The paragraph in question only provides an example of different datasets that might benefit from different noise characteristics.
>
> **Concern 4**: The introduction of auxiliary variables does not lower the posterior gap.
>
> Please note that our results demonstrate empirically that introducing both auxiliary variables and learned noise most likely __improves the posterior gap__: these methods yield significantly better ELBO values on CIFAR10 and ImageNet.
>
> However, if we were to only introduce auxiliary variables without adaptive noise, we would indeed get a worse posterior gap. This is confirmed by our ablation study (orange line in Figure 1a). However, this performance drop is more than compensated by learning the noise process (red line in Figure 1).

---

> ### Author Response · Authors · 2023-11-23
> **Response to reviewer U1Ra's questions**
>
> **Question 1**: How was the BPD calculated? Is it VLB or ODE-based? What is the variance of the BPD?
>
> We used both VLB and ODE-based methods to compute BPD.
>
> In the VLB-based approach, we employ Eqn. 9. We use T = 128 in Eqn. 10, discretizing the timesteps [0, 1] into 128 bins. For the ODE-based approach, we follow the exact evaluation procedure in i-DODE: we extract the underlying ODE for the diffusion process and calculate the likelihood using ODE-based exact methods. We also use their importance-weighted bound with K importance samples (note that K=1 corresponds to the ELBO, and K>1 provides a tighter bound on the likelihood).
>
> Below, we report BPD values (mean and 95% Confidence Interval) for MuLAN on CIFAR10 (8M training steps) and ImageNet (2M training steps) using both the VLB-based approach, and the ODE-based approach with K=1 and K-20 importance samples.
>
> | Approach | CIFAR10 BPD | ImageNet BPD |
> | --- | --- | --- |
> VLB-based  | 2.59 ± 1e-3 | 3.71 ± 1e-3
> ODE-based (K=1) | 2.59 ± 3e-4 | 3.71 ± 1e-3
> ODE-based (K=20) | 2.55 ± 3e-4 | 3.67 ± 1e-3 *
>
> \* *Partial result obtained on 70% of the test data due to time constraints.*
>
> **Question 2**: Did the authors analyze the auxiliary latent space? Was there any interpretable meaning?
>
> We analyzed the effects of the auxiliary variables on the noise process. While we did observe variation in the noise schedule for different auxiliary variables (Figure 2), we did not find any human-interpretable patterns. We defer the human-interpretable analysis to future work. We did not attempt to analyze the ability of the auxiliary variables to perform representation learning; however, that problem has been extensively studied by Wang et al., 2023.

---

### Official Review · Reviewer_oSn9 · 2023-11-01

**Soundness:** 2 fair
**Presentation:** 3 good
**Contribution:** 3 good
**Rating:** 5
**Confidence:** 4

**Summary:**

In this paper, the authors provided a theoretical argument that creating a noise schedule for each input dimension and conditioning it on the input yields improved likelihood estimation, which means noise with different covariance matrices can be applied to the inputs. Furthermore, the authors introduced a novel method to condition the noise schedule on the input via a latent distribution. Empirical experiments are made to demonstrate the effectiveness and efficiency of the new proposed model.

**Strengths:**

1. This paper is well written. The presentation is good and the reference list is complete.
2. The experiments in this paper is quite solid.

**Weaknesses:**

1. I recommend the authors show some generated images as well as the comparison with other existing models, so that we can see the improvement more clearly.
2. The theory proposed by the authors only showed us the pipeline of this model. For the reason why polynomial noise scheduling is better than the existing constant/linear/exponential noise scheduling still remains unclear. If it is difficult to obtain a solid theorem, I think it necessary to explain it more.
3. The idea proposed is not so impressive in my opinion, but it is not a serious weakness since the authors have done solid experiments and made the polynomial noise scheduling model come true.

**Questions:**

1. Do you use pretrained score estimator, or you trained your own? Since the polynomial noise scheduling is originally proposed by you, there are no pretrained score estimators to use I guess. Is it right?
2.There are no more additional questions. The authors only need to answer my questions in the "weakness" section.

---

> ### Author Response · Authors · 2023-11-23
> **Response to reviewer oSn9**
>
> We want to thank the reviewer for their constructive feedback. We address each concern below.
>
> **Concern 1**: Understanding the novelty of the paper relative to recent work.
>
> MuLAN is the **first** method to introduce a learned adaptive noise process. A widely held assumption is that the ELBO objective is invariant to the noise process (Kingma et al., 2021). We __dispel this assumption__: when input-conditioned noise is combined with (a) multivariate noise, (b) a novel polynomial parameterization, and (c) auxiliary variables, a learned noise process yields a tighter ELBO and a new state-of-the-art in density estimation. While (a), (c) were proposed in other contexts, we leverage them as subcomponents of a novel algorithm.
>
> The table below summarizes the relationship between MuLAN and other methods.
> | | Learned noise | Multivariate noise | Input Conditioned noise | Auxiliary latents | Noise parameterization |
> | --- | --- | --- | --- | ---  | --- |
> VDM (Kingma et al., 2021) | **Yes** | No |No | No | Monotonic neural network |
> Blurring Diffusion Model (Hoogeboom et al., 2022) | No | **Yes** | No | No | Frequency scaling |
> InfoDiffusion (Wang et al., 2023) | No | No | No | In denoising process | Cosine schedule |
> MuLAN (ours) | **Yes** | **Yes** | **Yes** | In noising & denoising processes | Polynomial, sigmoid
>
> **Concern 2**: The inclusion of samples from the model.
>
> We have added the samples to the paper in the appendix G.
>
> **Concern 3**: Explaining why a polynomial noise schedule performs empirically better
>
> The reason why a polynomial function works better than a sigmoid or a monotonic neural network as proposed by VDM is rooted in Occam’s razor. In Appendix D2, we show that a degree 5 polynomial is the simplest polynomial that satisfies several desirable properties, including monotonicity and having a derivative that equals zero exactly twice. Simpler noise processes (e.g., scalar, exponential) are not sufficiently expressive to achieve these properties. More expressive models (e.g., monotonic 3-layer MLPs) are more difficult to optimize, hence perform worse.
>
> **Question 1**: Were the models trained from scratch?
>
> Yes, all the models were trained from scratch.
>
> **Addendum**: New experimental results
>
> We report new experimental results, and we now achieve a new state-of-the-art in density estimation on CIFAR10 and ImageNet. The table below summarizes our results.
>
> | Method | CIFAR10 bpd | ImageNet bpd |
> |-------------|-------------|-------------|
> | PixelCNN (Van den Oord et al., 2016) | 3.03 | 3.83
> Image Transformer (Parmar et al., 2018) | 2.90 | 3.77
> Score SDE (Song et al., 2020) | 2.99 | -
> Improved DDPM (Nichol & Dhariwal, 2021) | 2.94 | -
> VDM (Kingma et al., 2021) | 2.65 | 3.72
> iDODE (Zheng et al., 2023) | 2.56  | 3.69
> MuLAN (ours; initial version) | 2.60 ± 1e-3 | 3.71 ± 1e-3
> MuLAN (**ours; rebuttal version; state-of-the-art**) | **2.55** ± 1e-3 | **3.67\*** ± 1e-3
>
>
> *\* Partial result obtained on 70% of the data due to time constraints.*
>
> Our updated method adds two architectural improvements: velocity reparameterization (as in Zheng et al., 2023) and truncated normal dequantization (Dinh et al., 2017; Salimans et al., 2017; Ho et al., 2019; Zheng et al., 2023). For both iDODE and MuLAN with velocity parameterization, we report an importance-sampled estimate of test NLL with K=20 samples, as in the iDODE paper (sec A.1., eqn. 32).

---

### Official Review · Reviewer_PeVN · 2023-11-06

**Soundness:** 3 good
**Presentation:** 3 good
**Contribution:** 3 good
**Rating:** 6
**Confidence:** 4

**Summary:**

The paper suggests a method for teaching diffusion models to adapt their noise schedules in order to increase the ELBO (Evidence Lower BOund). The authors observe that if the noise schedule is expanded to include multiple variables, the ELBO for diffusion models will change depending on the noise schedule. This variation allows for the noise schedule to be optimized at the same time as the diffusion model parameters to enhance likelihood. The authors also explore instance-conditional diffusion along with auxiliary variables. They discovered that using these multivariate noise schedules combined with auxiliary variables enables the training of diffusion models that not only surpass previous benchmarks in terms of likelihood but also converge more quickly.

**Strengths:**

* The paper under review brings to light that the Evidence Lower BOund (ELBO) for continuous-time diffusion models, as described in the Variational Diffusion Model (VDM) paper, remains unchanged across various noise schedules only when the noise is univariate. It presents a novel finding that for multivariate noise schedules, the ELBO transforms into a line integral and varies with different noise schedules. This is insightful and could pave the way for further research in diffusion models.

* While VDM sets a challenging benchmark in terms of log-likelihoods, the paper in question surpasses these results, which is very impressive.

* Although the use of auxiliary variables in diffusion models isn't a new concept, the approach of conditioning noise schedules on such variables, as shown in this paper, is a valuable contribution that enhances model likelihoods.

* The clarity of the writing and the effective presentation of the paper are commendable.

**Weaknesses:**

* The concept of multivariate noise schedules is intriguing; however, it appears that it does not function effectively by itself and requires auxiliary variables for better performance. This raises the question of whether the combined learning of noise schedules and the diffusion model is advantageous.

* The potential improvements for Variational Diffusion Models (VDM) through the use of auxiliary latent variables remain unclear. It would be helpful to understand the significance of these variables in enhancing the likelihood. Although the authors have provided ablation studies for MuLAN without multivariate aspects, it's uncertain whether this is directly comparable to VDM with an auxiliary variable due to possible differences in noise schedule parameterization.

* The authors justify the learning of noise schedules based on the manual adjustment of such schedules in high-resolution image diffusion models. Yet, they focus on maximizing the Evidence Lower Bound (ELBO) for their learning method, while current diffusion models tend to optimize a different objective that emphasizes perceptual quality of samples. Whether their method is applicable or beneficial when the goal is not ELBO is not clear.

* Additionally, the discussion of related works could be more comprehensive. The paper frequently refers to continuous-time diffusion models but often overlooks citation [1], from which such models originate. The authors could provide a more thorough background for readers by acknowledging concurrent works [2] and [3], which propose the same ELBO for continuous-time diffusion models. This inclusion would add value to the context in which VDM is discussed.

References:

[1] Song, Y., Sohl-Dickstein, J., Kingma, D.P., Kumar, A., Ermon, S. and Poole, B., 2020. Score-based generative modeling through stochastic differential equations. arXiv preprint arXiv:2011.13456.

[2] Song, Y., Durkan, C., Murray, I. and Ermon, S., 2021. Maximum likelihood training of score-based diffusion models. Advances in Neural Information Processing Systems, 34, pp.1415-1428.

[3] Huang, C.W., Lim, J.H. and Courville, A.C., 2021. A variational perspective on diffusion-based generative models and score matching. Advances in Neural Information Processing Systems, 34, pp.22863-22876.

**Questions:**

I would like to hear the authors' thoughts on the weaknesses identified above.

---

> ### Author Response · Authors · 2023-11-23
> **Response to reviewer PeVN**
>
> We want to thank the reviewer for their constructive feedback. We address each concern below.
>
> **Concern 1**: Multivariate noise requires auxiliary variables to be useful. This raises the question of whether the combined learning of noise schedules and the diffusion model is useful.
>
> The combined learning of the noise schedules and the diffusion model is beneficial, as illustrated in Fig. 1. Neither a multivariate noise schedule nor an auxiliary variable scalar noise schedule individually leads to improvement; however, when these are combined, MuLAN yields a significantly better BPD than VDM.
>
> Note also that we could not learn the noise separately from the diffusion model in MuLAN: they are coupled via the auxiliary variables, which are found in both.
>
> **Concern 2**: The improvement over VDMs through the use of auxiliary variables is unclear. What is the role of these variables in improving log-likelihood?
>
> Auxiliary variables by themselves do not improve the log-likelihood of VDMs: we show this in Figure 1. Auxiliary variables require applying the ELBO twice: this yields a training objective that bounds the marginal log-likelihood less tightly than in VDM. By itself, this lowers performance (Figure 1; orange line). However this performance drop is more than compensated by learning the noise process (Figure 1; red line). Thus, obtaining good log-likelihood requires using the entire set of components introduced in MuLAN (including noise that is multivariate, adaptive, and learned).
>
> **Concern 3**: The motivation comes from the manual adjustment of noise schedules in diffusion models. Does mulan lead to improved performance when the obj. Isn’t ELBO?
>
> The main motivation of our work comes from variational inference: we view the noising process as a variational posterior, and we learn it to obtain a tighter ELBO (see Section 3.1). This in turn yields state-of-the-art density estimation.
>
> Note that the manual adjustment of noising schedules in high-resolution diffusion models is not the motivation for our work. We also do not claim that likelihood correlates with image quality. The passage in question is only a citation to relevant concurrent work.
>
> Lastly, the goal of this paper is to study how noise schedule affects the likelihood. We leave to future work the question of whether a learnable noise schedule is also beneficial when the objective function isn’t ELBO. Since there is evidence that manual adjustment of the noise schedule helps with improving the perceptual quality of the images (Chen, 2023; Hoogeboom et al., 2023), methods inspired by MuLAN hold promise there.
>
> **Concern 4**: Lack of a citation to Song et al., 2020.
>
> We thank the reviewer for bringing this to our attention. We have added the reference to our paper.
>
> **Addendum**: New experimental results
>
> We report new experimental results, and we now achieve a new state-of-the-art in density estimation on CIFAR10 and ImageNet. The table below summarizes our results.
>
>
> | Method | CIFAR10 bpd | ImageNet bpd
> | --- | --- | --- |
> PixelCNN (Van den Oord et al., 2016) | 3.03 | 3.83
> Image Transformer (Parmar et al., 2018) | 2.90 | 3.77
> Score SDE (Song et al., 2020) | 2.99 | -
> Improved DDPM (Nichol & Dhariwal, 2021) | 2.94 | -
> VDM (Kingma et al., 2021) | 2.65 | 3.72
> iDODE (Zheng et al., 2023) | 2.56  | 3.69
> MuLAN (ours; initial version) | 2.60 ± 1e-3 | 3.71 ± 1e-3
> MuLAN (**ours; rebuttal version; state-of-the-art**) | **2.55** ± 1e-3 | **3.67\*** ± 1e-3
>
>
> *\* Partial result obtained on 70% of the data due to time constraints.*
>
> Our updated method adds two architectural improvements: velocity reparameterization (as in Zheng et al., 2023) and truncated normal dequantization (Dinh et al., 2017; Salimans et al., 2017; Ho et al., 2019; Zheng et al., 2023). For both iDODE and MuLAN with velocity parameterization, we report an importance-sampled estimate of test NLL with K=20 samples, as in the iDODE paper (sec A.1., eqn. 32).

---

### Author Response · Authors · 2023-11-23
**Response to all reviewers**

We thank the reviewers for their helpful feedback. Here is a summary of the new results we obtained in response to their concerns:
* **New experimental results.** We now report a new **state-of-the-art in density estimation** on CIFAR10 and ImageNet. We report a BPD of 2.55 on CIFAR10 and a BPD of 3.67 on ImageNet; both numbers outperform i-DODE (Zheng et al., 2023), which is the current best method.
* **Improved comparison to previous work.** We include a comparison to InfoDiffusion, VDM, and other models identified by the reviewers. We introduce a table that illustrates the key differences between each method.
* **Additional clarifications.** We clarified the motivation of our work, the role of individual components (multivariate noise, auxiliary variables, and we fixed several typos and references.

We appreciate the opportunity to improve our manuscript based on the reviewers' feedback and believe that incorporating these changes will enhance the quality and clarity of our work.

---

### Meta-Review · Area_Chair_THfg · 2023-12-08

**Metareview:**

The paper proposes a method for enhancing diffusion models by adapting noise schedules, optimizing parameters, including instance-conditional diffusion and auxiliary variables, resulting in models that surpass previous likelihood benchmarks and converge more quickly.

The reviewers' feedback indicates a moderate reception of the paper. One primary concern revolves around the significance of the improvement in likelihood for diffusion models. While acknowledging the potential importance of this metric for diffusion models and its correlation with downstream tasks such as visual quality (FID), mode covering (Precision and Recall), feature extraction using the encoded representation, or likelihood-based out-of-distribution detection, there is uncertainty about the standalone utility of BPD in the context of diffusion models, which differ from typical VAEs.

Although the introduction of a learnable adaptive noise process is acknowledged as beneficial for reducing BPD, there remains skepticism about the absence of other relevant metrics, which, in the AC's opinion, should not exhibit a clear decline.

**Justification For Why Not Higher Score:**

The paper exhibits a misalignment between the justification for learning noise schedules in high-resolution image diffusion models and its focus on maximizing the Evidence Lower Bound (ELBO), prompting questions about the applicability of the proposed approach in scenarios prioritizing perceptual quality over ELBO. Furthermore, the experiment results, though significant compared to some methods, are perceived as underwhelming relative to the original Variational Diffusion Models (VDM), casting doubt on the effectiveness and novelty of the proposed method.

**Justification For Why Not Lower Score:**

N/A

---

### Decision · Program_Chairs · 2024-01-16

Reject